# Nuclear-resident RIG-I senses viral replication inducing antiviral immunity

GuanQun Liu [1,2], Yao Lu[1,3], Sathya N. Thulasi Raman[1], Fang Xu[1], Qi Wu[1,3], Zhubing Li[1,2], Robert Brownlie[1], Qiang Liu [1,2,3] & Yan Zhou [1,2,3]

The nucleus represents a cellular compartment where the discrimination of self from non-self nucleic acids is vital. While emerging evidence establishes a nuclear non-self DNA sensing paradigm, the nuclear sensing of non-self RNA, such as that from nuclear-replicating RNA viruses, remains unexplored. Here, we report the identification of nuclear-resident RIG-I actively involved in nuclear viral RNA sensing. The nuclear RIG-I, along with its cytoplasmic counterpart, senses influenza A virus (IAV) nuclear replication leading to a cooperative induction of type I interferon response. Its activation signals through the canonical signaling axis and establishes an effective antiviral state restricting IAV replication. The exclusive signaling specificity conferred by nuclear RIG-I is reinforced by its inability to sense cytoplasmic-replicating Sendai virus and appreciable sensing of hepatitis B virus pregenomic RNA in the nucleus. These results refine the RNA sensing paradigm for nuclear-replicating viruses and reveal a previously unrecognized subcellular milieu for RIG-I-like receptor sensing.

---

[1] Vaccine and Infectious Disease Organization-International Vaccine Centre (VIDO-InterVac), University of Saskatchewan, Saskatoon, SK S7N 5E3, Canada. [2] Vaccinology & Immunotherapeutics Program, School of Public Health, University of Saskatchewan, Saskatoon, SK S7N 5E3, Canada. [3] Department of Veterinary Microbiology, Western College of Veterinary Medicine, University of Saskatchewan, Saskatoon, SK S7N 5E3, Canada. Correspondence and requests for materials should be addressed to Y.Z. (email: yan.zhou@usask.ca)

The constant challenge of the vertebrate cells by invading pathogens drives the evolution of innate immune systems to rapidly detect and respond to non-self molecules, such as the virus-derived nucleic acids[1]. Depending on their intracellular localization, distinct germline-encoded pattern-recognition receptors (PRRs) engage specialized adapters to initiate immune signaling cascades from within different cellular compartments. To date, it has been well defined the roles of PRRs, including the Toll-like receptors, retinoic acid inducible gene I (RIG-I)-like receptors (RLRs), and cyclic GMP-AMP synthase (cGAS), in endosomal and cytosolic sensing of viral DNA and RNA[2,3]. However, the nuclear-replicating property of nearly all DNA viruses has shifted the PRR sensing paradigm toward the cell nucleus[4]. Emerging evidence delineates the functional roles of DNA sensors including IFI16 and cGAS in nuclear sensing of the *Herpesviridae* family, including Herpes simplex virus 1, human cytomegalovirus, Epstein-Barr virus, and Kaposi sarcoma-associated virus[5–8]. In contrast, most RNA viruses replicate within the cytoplasmic compartment. One of the primary RNA sensors, RIG-I, is well characterized as a cytosolic sensor of viral RNAs bearing 5′ diphosphates or triphosphates and juxtaposed short base-paired stretches[9]. Its activation initiates signaling cascades via the mitochondrial antiviral-signaling protein (MAVS), leading to the production of type I interferons (IFNs) which in turn upregulate other antiviral interferon-stimulated genes (ISGs)[10,11]. Interestingly, the members of the *Orthomyxoviridae* family represent a few RNA viruses replicating in the nuclear compartment[12]; however, the existence of a nuclear RNA sensing paradigm remains unexplored.

Apart from the IFN antagonism exerted by a plethora of virus-encoded IFN antagonistic proteins targeting IFN induction and IFN signaling axes[13,14], virus-mediated compartmentalization has recently become another immune evasion strategy. The members of the *Flaviviridae* family, including tick-borne encephalitis virus and hepatitis C virus, induce the formation of compartmentalized membrane structures to sterically segregate their viral agonists from PRRs[15,16]. This strategy conceals the viral replication site from the cytosolic RLRs, thereby minimizing the likelihood of non-self RNA sensing. Likewise, the nuclear replication of the *Orthomyxoviridae* family has also been hypothesized to act as an immune evasion strategy due in part to the cytoplasmic localization of classical RNA sensors such as RIG-I[17,18]; however, this has never been experimentally substantiated. On the other hand, the host may have evolved a nuclear RNA sensing mechanism to counteract such immune evasion.

Influenza A virus (IAV) is the most-studied member of the *Orthomyxoviridae* family whose genome transcription and replication are closely associated with nuclear machinery[19]. It stimulates type I IFN expression in infected cells via a nearly strict RIG-I-dependent signaling cascade[11,20]. This RIG-I dependency underscores a long-standing question as to how the cytosolic RNA sensor RIG-I senses the nuclear replicating IAV. To date, the only RIG-I agonist characterized for influenza virus is the panhandle structure residing in either full-length or defective-interfering (DI) viral genomes[21–23]. We also reported that the IAV panhandle structure mediates and is mainly responsible for RIG-I activation and IFN induction in vitro[24]. Nonetheless, the spatiotemporal detection of the panhandle structure by RIG-I during the course of IAV infection remains unknown, particularly the accessibility of the ligands to RIG-I given the nuclear replication nature of IAV. Although an apparent interaction of cytoplasmic RIG-I with incoming viral ribonucleoprotein (vRNP) was visualized, no clear correlation with IFN induction was observed[25,26]. Moreover, IAV differs from influenza B virus (IBV) in the kinetics of IFN induction; while IBV activates IFN signaling immediately after infection, IAV evades early recognition and

induces IFN production at the late stage of infection[27,28]. Furthermore, IFN induction upon IAV infection could not be detected when viral RNA synthesis is inhibited[29]. These observations strongly suggest that RIG-I is capable of sensing IAV during the course of its nuclear replication, though the nature of the viral RNA species recognized remains unclear.

Here, we investigate the spatiotemporal activation of RIG-I in relation to IFN induction during IAV infection and refine the RIG-I sensing paradigm for IAV. We identify nuclear-resident RIG-I that is actively involved in sensing IAV replication in the nucleus, resulting in a cooperative IFN induction along with its cytoplasmic counterpart. Nuclear RIG-I initiates a MAVS-dependent canonical signaling cascade, and is effective in establishing an antiviral state that restricts IAV infection. Furthermore, nuclear RIG-I remains inactive upon infection with cytoplasmic-replicating Sendai virus (SeV), but exhibits signaling specificity toward nucleus-derived viral agonists, such as the pregenomic RNA (pgRNA) of hepatitis B virus (HBV). Our findings unravel a previously unrecognized nuclear pool of RIG-I sensing nuclear viral RNA and implicate a novel subcellular milieu for RLR sensing.

## Results

**Genuine presence of nuclear-resident RIG-I.** RNA sensors are among the core vertebrate ISGs which exhibit very low basal expression but are highly upregulated by IFN[30]. In a focused exploration for the nuclear existence of the otherwise cytoplasmic RNA sensor RIG-I, we subjected A549 cells to SeV infection or IFNβ stimulation to globally upregulate RIG-I levels. The subcellular distribution of RIG-I was examined by immunofluorescence and subcellular fractionation. To ensure any observed RIG-I localization by immunofluorescence was genuine, *RIG-I* KO A549 cells were generated by CRISPR/Cas9-mediated genome editing (Supplementary Fig. 1) and the specificity of RIG-I antibodies from different sources was carefully monitored by observing RIG-I staining in *RIG-I* KO A549 cells stimulated with SeV (Supplementary Fig. 2a). In the course of infection, SeV stimulated increasing amount of endogenous RIG-I expression, concomitant with increased levels of interferon regulatory factor 3 (IRF3) nuclear translocation (Fig. 1a and Supplementary Fig. 2b). Similarly, significant upregulation of RIG-I expression was observed in IFNβ-primed cells (Supplementary Fig. 2b). Noticeably, a distinct nuclear staining of RIG-I was consistently observed, albeit at a much-reduced level compared with cytoplasmic RIG-I (Fig. 1a and Supplementary Fig. 2b). In THP-1 cells and HeLa cells ectopically expressing a FLAG-tagged RIG-I, nuclear RIG-I staining was also detected using either RIG-I antibody or a highly specific monoclonal antibody against the FLAG epitope (Supplementary Fig. 2c, d). To further authenticate the microscopic observation, SeV-infected or IFNβ-primed A549 cells were subjected to cellular fractionation and the presence of nuclear RIG-I was consistently revealed (Fig. 1b, c, lanes 1–4). In addition, we examined whether the nuclear localization of RIG-I requires its downstream mitochondrial adapter MAVS. To this end, *MAVS* KO A549 cells were generated which could be efficiently primed by IFNβ for RIG-I upregulation (Fig. 1c, lanes 5–8), but were abolished in SeV-induced IRF3 phosphorylation and RIG-I upregulation due to a lack of MAVS-dependent IFN induction (Supplementary Fig. 2e). As with that in wild-type (WT) A549 cells, nuclear RIG-I was also detected in *MAVS* KO cells primed with IFNβ (Fig. 1c), demonstrating that the nuclear localization of RIG-I was independent of MAVS.

To further corroborate the presence of RIG-I in the nucleus, RIG-I was expressed with a FLAG tag together with a well-established nuclear export signal (NES)[31]. We asked whether the

NES would redirect RIG-I out of the nucleus and negate the observed nuclear localization. To obviate potential interference from endogenous RIG-I, *RIG-I* KO A549 cells were complemented with stably maintained episomal constructs inducibly expressing a scrambled- (SLN-) or NES-tagged RIG-I. In response to doxycycline (Dox), these cells expressed high levels of respective RIG-I proteins (Fig. 1d). In contrast to SLN-RIG-I, which showed consistently detectable nuclear staining, NES-RIG-I exhibited negligible nuclear localization (Fig. 1d). These cells

were subsequently infected with IAV to examine whether virus infection induces RIG-I redistribution. Interestingly, IAV infection did not appear to change RIG-I localization, as both SLN-RIG-I and NES-RIG-I exhibited the similar cellular distribution in infected cells compared to mock infection (Fig. 1d, e). Pearson's correlation coefficient analysis further confirmed a significantly greater colocalization of SLN-RIG-I with DAPI staining in the nucleus than that of NES-RIG-I (Fig. 1f). As a control, colocalization between nuclear IAV proteins and DAPI

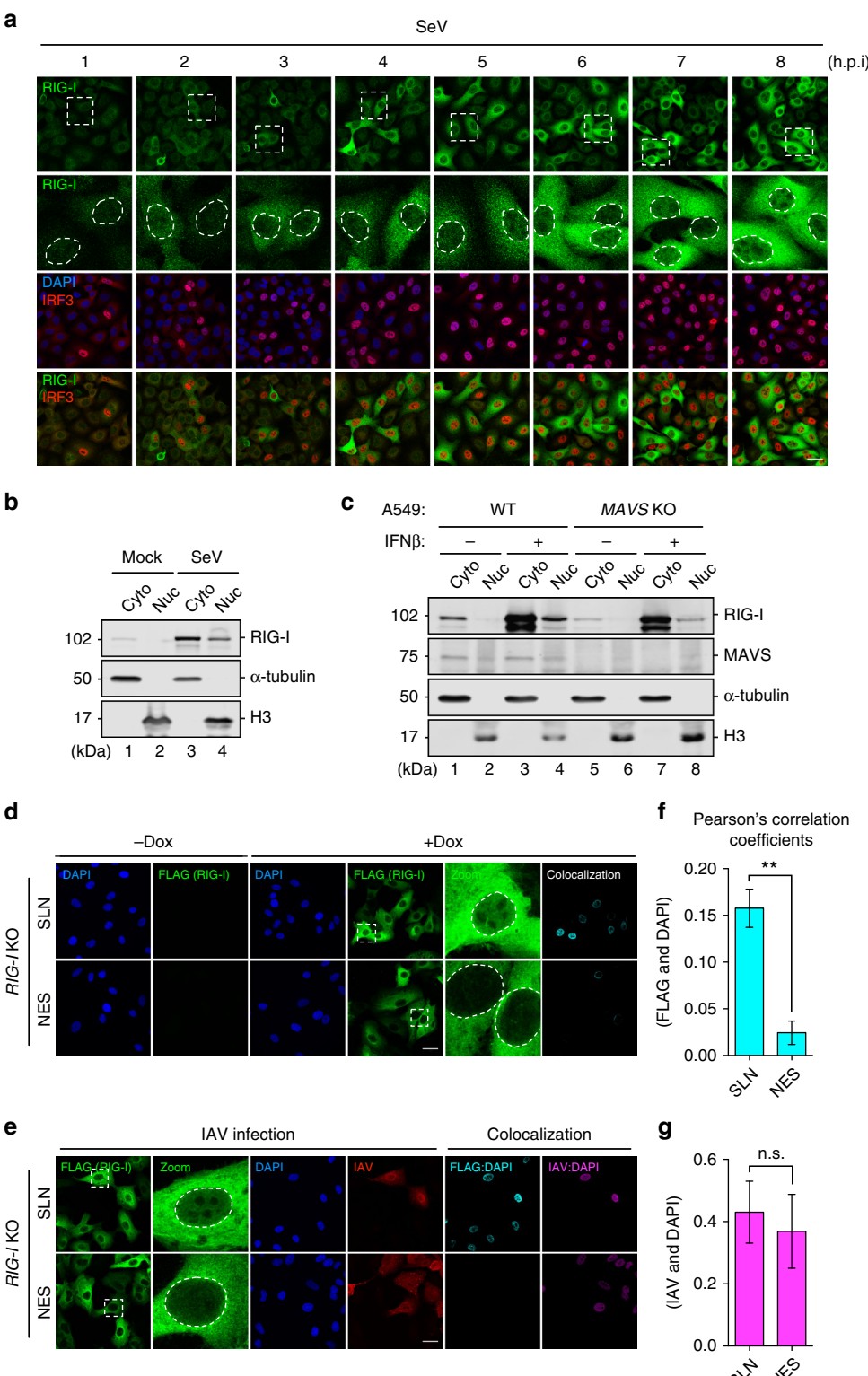

staining remained unchanged in infected cells expressing either SLN-RIG-I or NES-RIG-I (Fig. 1g). Collectively, these results demonstrated the genuine presence of RIG-I in the nucleus, which prompted us to further explore a plausible contribution of nuclear RIG-I to the sensing of nuclear viral RNA, particularly that of nuclear-replicating IAV.

**Nuclear RIG-I constantly binds IAV vRNPs in the nucleus.** RIG-I activation is generally inhibited during WT IAV infection unless the major IFN antagonistic protein NS1 is disrupted[32,33]. In contrast to that observed during SeV infection (Fig. 1a), endogenous RIG-I expression remained at basal levels in A549 cells infected with NS1-deficient IAV (ΔNS1-ms) during a single-cycle infection (Fig. 2a). Upon prolonged infection, endogenous RIG-I levels remained low in the infected cells, whereas RIG-I upregulation was evident in non-infected neighboring cells, indicating an antiviral state primed by the IFN produced from the infected cells (Supplementary Fig. 3a). Since the low level of RIG-I in IAV-infected cells greatly hampered the tracking of its distribution by immunofluorescence, we sought to generate a recombinant mutant virus deficient of both NS1 and another viral protein PA-X which poses a strong shutoff activity that globally supresses host protein expression[34]. We were able to rescue a double-deficient virus (PAfsΔNS1), whose PA and NS segments carry X-ORF frameshifting mutations and NS1-ORF deletion, respectively. PAfsΔNS1 was significantly attenuated in growth in IFN competent cells but replicated to a level that was two-log lower than the WT and ΔNS1-ms viruses in MDCK cells expressing NS1. Although no greater RIG-I upregulation was observed in infected cells during a single-cycle (Fig. 2b) or prolonged infection (Supplementary Fig. 3a), PAfsΔNS1 stimulated IRF3 nuclear translocation as efficiently as ΔNS1-ms (Fig. 2a, b). Of note, IRF3 nuclear translocation was abolished in *RIG-I* KO A549 cells infected with either virus, demonstrating a strict activation of the RIG-I signaling pathway by both viruses (Supplementary Fig. 3b). Interestingly, a comparison between the two RIG-I-activating mutant viruses revealed distinct patterns of vRNP distribution. Using either fluorescence in situ hybridization (FISH) specifically detecting the M segment vRNA or the viral NP protein as a vRNP indicator, we observed substantial vRNP retention for PAfsΔNS1 in the nucleus during a single-cycle infection, whereas ΔNS1-ms exhibited vRNP distribution in both the nucleus and cytoplasm (Fig. 2a, b and Supplementary Fig. 3b). Despite such significant difference in vRNP localization, the kinetics of IRF3 activation induced by the two viruses resembled each other (Fig. 2c). Moreover, monitoring the vRNA localization in relation to IRF3 nuclear translocation revealed that RIG-I activation was closely associated with the nuclear vRNA accumulation; IRF3 activation was first detectable at 4 h.p.i and increased up to 8 h.p.i, concomitant with the course of nuclear vRNA accumulation of PAfsΔNS1 (Fig. 2b). Taken together, that

PAfsΔNS1 manifested impaired vRNP nuclear export yet stimulated IRF3 nuclear translocation as efficiently as ΔNS1-ms indicated a nuclear sensing event by the nuclear-resident RIG-I.

To sense nuclear viral RNA synthesis, the nuclear RIG-I would possibly interact with IAV vRNP to gain closer proximity to the immunostimulatory RNA species, such as the vRNA and cRNA[21,24]. We performed cellular fractionation followed by co-immunoprecipitation to examine whether nuclear RIG-I associates with vRNPs. During WT or ΔNS1-ms infection, endogenous nuclear RIG-I interacted with vRNPs as efficiently as cytoplasmic RIG-I (Fig. 2d). Given the relative amount of immunoprecipitated proteins, nuclear RIG-I associated with even more vRNPs than the cytoplasmic RIG-I (Fig. 2d). Nuclear RIG-I association with vRNPs was likewise detected for seasonal H3N2 (A/Victoria/3/75) and avian H7N3 (A/chicken/British Columbia/CN-6/2004) viruses (Fig. 2e), ruling out a subtype-specific effect. To evaluate the relative temporal contribution of the two cellular pools of RIG-I to vRNP association, we monitored the extent of vRNP binding by cytoplasmic and nuclear RIG-I at early and late time points after WT virus infection. While cytoplasmic RIG-I associated with limited levels of vRNPs at 6 h.p.i, the interaction between nuclear RIG-I and vRNPs was noticeably higher (Fig. 2f, lane 3 vs. lane 6). At 14 h.p.i, the cytoplasmic RIG-I showed increased levels of vRNP association compared to that at 6 h.p.i (Fig. 2f, lane 3 vs. lane 9), while nuclear RIG-I manifested constant vRNP association (Fig. 2f, lane 6 vs. lane 12). These observations fit in the time frame of IAV life cycle during which the incoming vRNPs are rapidly imported into the nucleus where the nuclear RIG-I accounts for the early recognition. With the increased nuclear export of progeny vRNPs, both pools of RIG-I are involved in vRNP interaction at the late stage of infection. Collectively, these results demonstrated a sustained interaction of nuclear-resident RIG-I with vRNP in the IAV life cycle, particularly at early time points when the abundance of cytoplasmic RIG-I agonists is limited.

**Nuclear-localized RIG-I senses nuclear viral RNA replication.** To further scrutinize the functional role of nuclear RIG-I in RNA sensing, we took advantage of the IAV RNP reconstitution system, which recapitulates nuclear RNA synthesis with the minimal set of viral proteins[35]. Consistent with that in infected A549 cells (Fig. 2d), a nuclear pool of RIG-I was detected in RNP reconstituted 293T cells ectopically expressing a FLAG-tagged RIG-I which also bound to nuclear vRNPs (Supplementary Fig. 4a). The RNA species extracted from the nuclear RIG-I immunoprecipitates were immunostimulatory when transfected into 293T reporter cells (Supplementary Fig. 4b). We next examined the cellular localization of viral RNA species in RNP reconstituted cells, particularly that of the essential RIG-I agonists (vRNA and cRNA) as previously reported[21,24]. While a portion of viral mRNA was exported to the cytoplasm, full-length vRNA and

**Fig. 1** Genuine presence of RIG-I in the cell nucleus. **a** A549 cells were infected with Sendai virus (SeV) at 50 HAU/mL. At indicated time points post-infection, cells were subjected to immunofluorescence for RIG-I (green) and IRF3 (red). Nuclei were stained with DAPI (blue). Boxed area was enlarged and the nuclear regions were outlined to highlight the nuclear RIG-I staining. Scale bar = 25 μm. **b** A549 cells were left uninfected, or infected with SeV (50 HAU/mL) for 16 h. Cells were subjected to cellular fractionation and the presence of RIG-I in the cytoplasmic (Cyto) and nuclear (Nuc) fractions was determined by immunoblotting. α-tubulin and histone H3 served as the markers for the cytoplasmic and nuclear fractions, respectively. **c** A549 WT or *MAVS* KO cells were left untreated, or primed with human IFNβ (500 U/mL) for 16 h. Cellular fractionation was performed as in **b**, and the presence of RIG-I and MAVS in both fractions was determined by immunoblotting. **d, e** A549 *RIG-I* KO cell lines inducibly expressing SLN-RIG-I or NES-RIG-I were left non-induced (−Dox), or induced with 1 μg/mL doxycycline (+Dox) for 4 h followed by mock (**d**) or WT PR8 infection (**e**, MOI = 5) for 14 h. Cells were subjected to immunofluorescence for FLAG-RIG-I (green) and IAV (red). Nuclei were stained with DAPI (blue). Scale bar = 25 μm. Colocalization of FLAG/DAPI channels, and IAV/DAPI channels was analyzed using ImageJ and the colocalizing pixels were represented in cyan and magenta, respectively. **f, g** Pearson's correlation coefficients were calculated from at least three random fields with ~20 cells. Data are shown as mean ± SD. Significant differences were determined by an unpaired Student's *t*-test. **p < 0.01; n.s. not significant

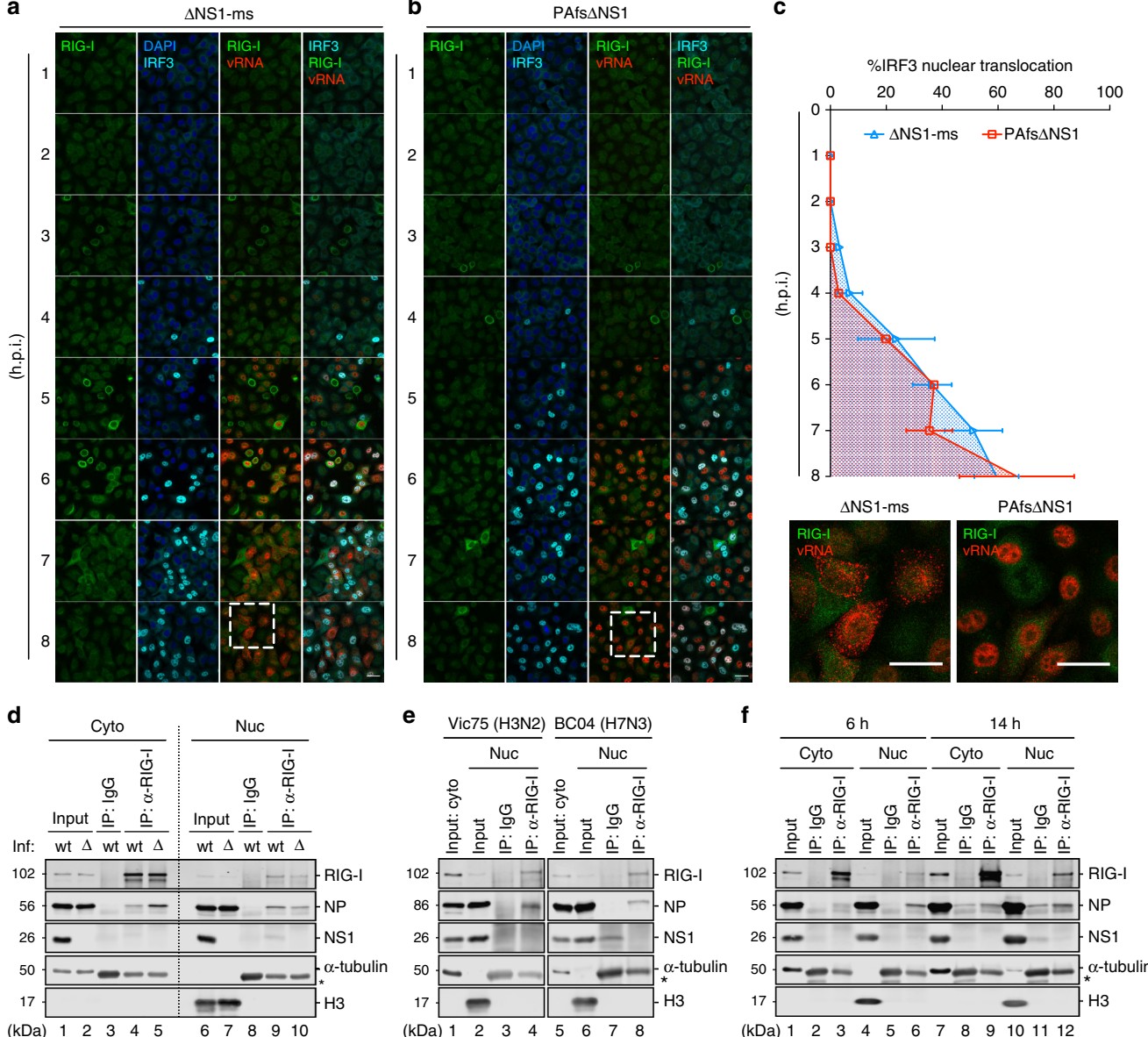

**Fig. 2** Nuclear-resident RIG-I constantly binds IAV vRNPs in the nucleus. **a**, **b** A549 cells were infected with ΔNS1-ms or PAfsΔNS1 virus at an MOI of 10. At indicated time points post-infection, cells were subjected to immunofluorescence for RIG-I (green) and IRF3 (cyan), and FISH for vRNA of the M segment (red). Nuclei were stained with DAPI (blue). The insets highlight the distinct vRNA distribution of the two mutant viruses. Scale bar = 25 μm. **c** Percentage of A549 cells with IRF3 nuclear translocation upon ΔNS1-ms and PAfsΔNS1 infections was quantified from at least three random fields with ~100 cells for each time point after infection. **d** A549 cells were infected with WT PR8 or ΔNS1-ms (Δ) at an MOI of 5. At 8 h.p.i, cells were subjected to cellular fractionation and both cytoplasmic (Cyto) and nuclear (Nuc) fractions were immunoprecipitated (IP) with the RIG-I antibody or an IgG isotype control. The immunoprecipitates were analyzed by immunoblotting for RIG-I, NP, and NS1. α-tubulin and histone H3 served as the markers for the cytoplasmic and nuclear fractions, respectively. The heavy chains of IP antibodies are indicated by an asterisk (*). **e** Co-immunoprecipitation in the nuclear fractions of A549 cells infected with A/Victoria/3/75 (H3N2) or A/chicken/British Columbia/CN-6/2004 (H7N3) was performed as in (**d**). **f** A549 cells were infected with WT PR8 (MOI = 2) for 6 or 14 h. Cells were subjected to cellular fractionation and co-immunoprecipitation in both the cytoplasmic and nuclear fractions was performed as in (**d**)

cRNA of the NA segment were strictly confined to the nuclei of reconstituted cells (Fig. 3a). Reconstitution in the presence of a catalytically inactive PB1 (PB1a) served as the baseline for quantification of viral RNA levels[36]. Similarly, the FISH analysis revealed exclusive nuclear staining of M vRNA upon M segment reconstitution (Fig. 3b).

Accordingly, we specifically examined whether a nuclear-localized RIG-I responds to IAV RNA synthesis leading to IFN induction. RIG-I was tagged with an SV40 large T antigen nuclear localization signal (NLS) to direct it to the nucleus (Fig. 3b).

While SLN-tagged and NES-tagged RIG-I responded efficiently to the cytoplasmic delivery of a synthetic RIG-I ligand, NLS-RIG-I remained inactive upon stimulation (Fig. 3c). Next, 293T cells were reconstituted with each of the eight viral segments in the presence of SLN-RIG-I, NLS-RIG-I, or GFP. None of these constructs affected IAV polymerase activity (Supplementary Fig. 5a). Although SLN-RIG-I overexpression resulted in a detectable residual level of IFNβ promoter activation, reconstitution of the PA, HA, NA, and M segments significantly enhanced IFN induction (Fig. 3d). In contrast, NLS-RIG-I itself showed

minimal IFN induction but responded significantly to the reconstitution of most segments, except for the NP and NS segments (Fig. 3e). This IFN suppression by NP and NS reconstitutions was also observed in the presence of SLN-RIG-I (Fig. 3d). While the expression of NS1 conferred the inhibitory

effect, NP protein alone did not impede dsRNA-induced IFN induction (Supplementary Fig. 5b). However, IFNβ promoter activation by NA reconstitution was suppressed by NP expression in a dose-dependent manner (Supplementary Fig. 5c), suggesting that NP level negatively regulates the generation of

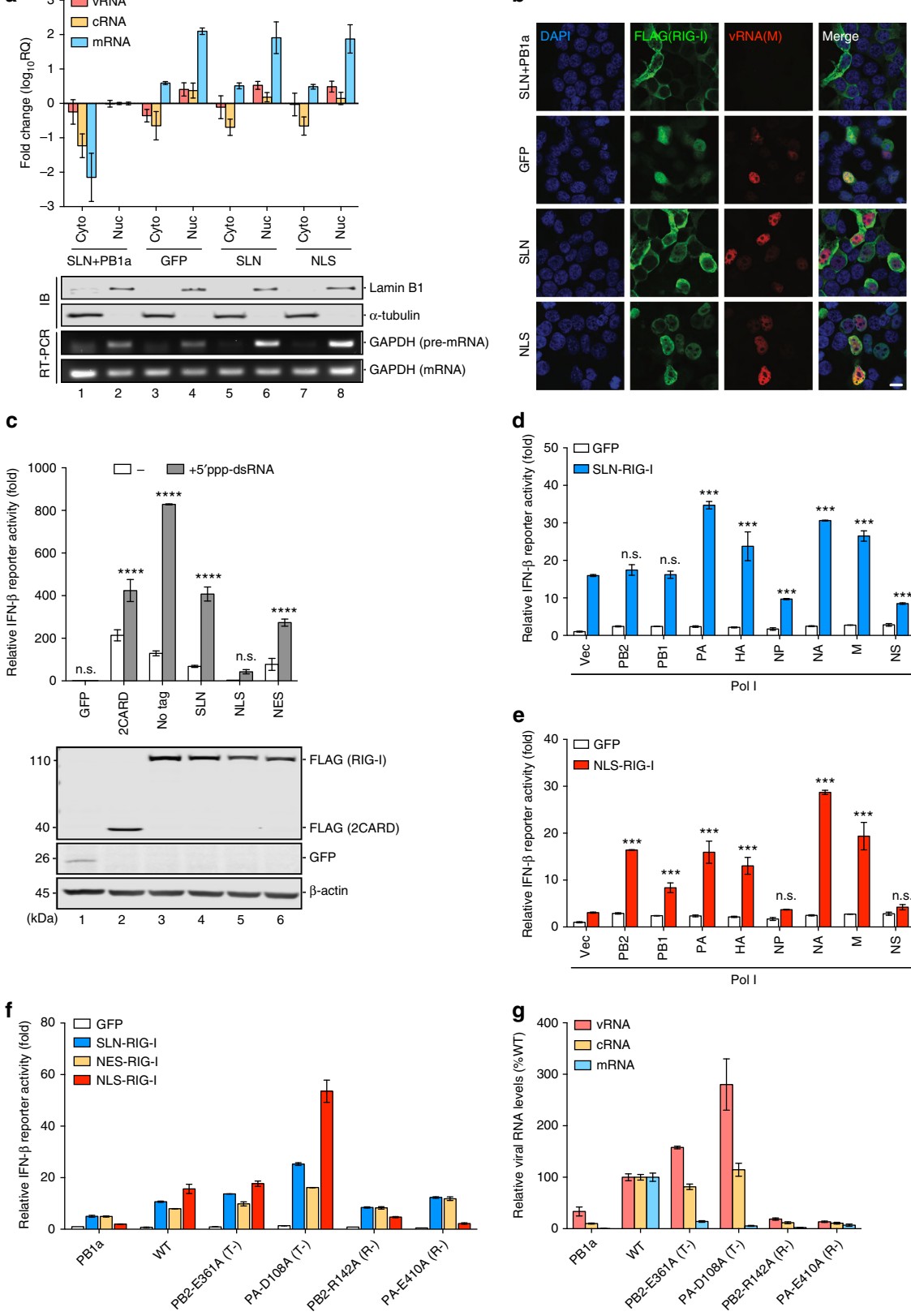

immunostimulatory RNA. Nonetheless, these results demonstrated that NLS-RIG-I was competent in sensing viral RNA synthesis in the nucleus.

Next, we examined the requirement of viral transcription and replication for NLS-RIG-I activation. Reconstitution in the presence of transcription-defective polymerases (PB2-E361A and PA-D108A)[37,38] was able to generate viral RNA species stimulating NLS-RIG-I. In contrast, no apparent IFNβ promoter activation was detected for reconstitution with replication-defective polymerases (PB2-R142A and PA-E410A)[39,40] (Fig. 3f). All mutant polymerases expressed to similar levels as the WT polymerase (Supplementary Fig. 6a). Quantification of viral RNA levels produced by mutant polymerases further revealed that the levels of IFNβ promoter activation were proportional to that of the full-length vRNA (Fig. 3f, g). This correlation was further confirmed by a dose-dependent activation of IFNβ promoter with increasing amount of vRNA-expressing plasmids (Supplementary Fig. 5d). Notably, compared to SLN-RIG-I and NES-RIG-I which consistently showed basal levels of activity, NLS-RIG-I exhibited superior signaling capacity in sensing augmented viral replication associated with transcription-defective polymerases (Fig. 3f, g). It is noted that no DI RNA was detected upon NA reconstitution (Supplementary Fig. 6b). Collectively, these results demonstrated that the nuclear-localized RIG-I senses nuclear viral RNA replication and is able to elicit greater IFN response possibly by gaining closer proximity to immunostimulatory viral RNA.

**Nuclear RIG-I senses IAV via the canonical signaling axis.** In the context of viral infection, we assessed the ability of nuclear RIG-I to sense IAV replication. *RIG-I* KO A549 cell lines inducibly expressing NLS-RIG-I and RIG-I with K270A mutation were constructed, with the latter serving as a signaling-inactive control[41]. As the ectopic expression of RIG-I for an extended period of time elicits autonomous IFN response thereby inhibiting viral infection, we limited the Dox-induced RIG-I expression in inducible A549 cells to 4 h. These cell lines, alongside with the SLN-RIG-I and NES-RIG-I-expressing cell lines, were subsequently infected with PAfsΔNS1 or ΔNS1-ms and probed for IRF3 nuclear translocation. Consistent with that in RNP-reconstituted 293T cells, complementation of *RIG-I* KO A549 cells with NLS-RIG-I mediated IRF3 nuclear translocation as efficiently as the SLN-RIG-I and NES-RIG-I, whereas K270A-RIG-I did not respond to IAV infection (Fig. 4a). Although PAfsΔNS1 exhibited confined vRNA to the nucleus, it elicited comparable levels of IRF3 activation to that of ΔNS1-ms infection (Fig. 4a). Given the low levels of endogenous nuclear RIG-I in WT cells, we titrated Dox concentration on NLS-RIG-I-expressing cells to match its expression comparable to the physiological levels (Supplementary Fig. 7a). Upon ΔNS1-ms

infection, low levels of NLS-RIG-I expression mediated efficient IFNβ and IP10 induction in a Dox dose-dependent manner (Supplementary Fig. 7b, c). Strikingly, compared to *RIG-I* KO cells, even trace amount of NLS-RIG-I remained active in sensing IAV infection and inducing antiviral gene expression (Supplementary Fig. 7d–g). Next, we compared the kinetics of IRF3 activation mediated by compartment-specific RIG-I in response to PAfsΔNS1 virus infection. Although IRF3 activation in cells expressing either RIG-I peaked at 7 h.p.i, both NES-RIG-I and NLS-RIG-I-expressing cell lines exhibited delayed IRF3 activation during 4 to 6 h.p.i (Fig. 4b, c), suggesting cooperative roles of cytoplasmic and nuclear RIG-I in sensing IAV replication.

We next delineated the signaling cascade involved in nuclear RIG-I sensing. In RNP reconstitution, we validated that NLS-RIG-I activation as determined by IFNβ promoter activation correlated with IRF3 phosphorylation and expression of antiviral genes such as IFNβ and IP10 (Fig. 4d and Supplementary Fig. 8). Point mutations that abrogated three well-characterized RIG-I signaling motifs were introduced into NLS-RIG-I[41–44]. NA reconstitution in the presence of NLS-RIG-I with abolished ATPase activity (K270A), tripartite motif-containing protein 25 (TRIM25) interaction (T55I), or RNA binding (K888E) failed to induce IFNβ promoter activation (Fig. 4e). Knockdown of MAVS by siRNA diminished NLS-RIG-I-mediated IFNβ promoter activation by NA reconstitution in 293T cells (Fig. 4f) and IRF3 phosphorylation in NLS-RIG-I expressing A549 cells infected with PAfsΔNS1 (Fig. 4g, lane 15 vs. lane 18). To further demonstrate the requirement of MAVS for nuclear RIG-I sensing, *RIG-I-MAVS* double KO (DKO) A549 cell line was generated followed by episomal complementation with NLS-RIG-I (Supplementary Fig. 9a). While *RIG-I* KO cells expressing NLS-RIG-I showed substantial IRF3 nuclear translocation upon PAfsΔNS1 infection, ablation of MAVS abolished NLS-RIG-I-mediated IRF3 activation in *RIG-I-MAVS* DKO cells (Fig. 4h). Moreover, no apparent change in the subcellular localization of MAVS was observed in infected cells expressing compartment-specific RIG-I (Supplementary Fig. 9b). These results demonstrated that the signaling pathway for NLS-RIG-I is consistent with the canonical RIG-I signaling axis for viral RNA sensing and IFN induction.

**Nuclear RIG-I elicits antiviral response restricting IAV.** We next sought to assess the antiviral effect conferred by nuclear RIG-I. The ΔNS1-ms virus was chosen to monitor the antiviral response as it had a relatively complete viral life cycle unlike the PAfsΔNS1 virus. At 14 h.p.i, ΔNS1-ms induced IFNβ and IP10 mRNA expression in *RIG-I* KO A549 cells expressing NLS-RIG-I, albeit to slightly lower levels than that in SLN-RIG-I and NES-RIG-I-expressing cells (Fig. 5a, b). Furthermore, NLS-RIG-I impeded viral protein accumulation as efficiently as SLN-RIG-I

**Fig. 3** Nuclear RIG-I senses nuclear viral replication during RNP reconstitution. **a** 293T cells were RNP reconstituted (PR8 NA segment) in the presence of GFP, SLN-RIG-I, or NLS-RIG-I for 24 h and were subjected to cellular fractionation. The levels of viral RNA species in the cytoplasmic (Cyto) and nuclear (Nuc) fractions were determined by strand-specific qRT-PCR. Fold change is expressed using the ΔΔCt method relative to the nuclear RNA fraction of the PB1a reconstitution. **b** RNP reconstituted 293T cells (PR8 M segment) were subjected to immunofluorescence for FLAG-RIG-I (green) and FISH for M vRNA (red) at 24 h.p.t. Nuclei were stained with DAPI (blue). Scale bar = 10 μm. **c** 293T cells were cotransfected with plasmids encoding GFP or various RIG-I constructs along with p125Luc and pTK-rLuc followed by transfection with a 19-bp 5′ppp-dsRNA. RLUs are expressed as fold change relative to the GFP group without RNA stimulation (lane 1). Expression levels of FLAG-RIG-I/2CARD and GFP were determined by immunoblotting. Significant differences between mock-transfected and dsRNA-transfected cells expressing GFP or either RIG-I constructs were determined by two-way ANOVA followed by Sidak post-test. **d**, **e** 293T cells were RNP reconstituted with a Pol I vector or each of the eight viral segments in the presence of GFP, SLN-RIG-I, or NLS-RIG-I for 24 h. RLUs are expressed as fold change relative to the Pol I vector reconstitution with GFP. Significant differences between reconstitutions with viral segments and Pol I vector were determined by one-way ANOVA followed by Tukey post-test. **f** 293T cells were RNP reconstituted (PR8 NA segment) using either inactive (PB1a), WT, transcription-defective (T-), or replication-defective (R-) polymerases in the presence of GFP, SLN-RIG-I, NES-RIG-I, or NLS-RIG-I for 24 h. RLUs are expressed as fold change relative to the PB1a reconstitution in the presence of GFP. **g** The levels of viral RNA species in RNP reconstituted cells (**f**) were determined by strand-specific qRT-PCR and are expressed as percentage changes compared to the reconstitution using WT polymerase. Data are shown as mean ± SD of three experiments. \*\*\*$p < 0.001$; \*\*\*\*$p < 0.0001$; n.s. not significant

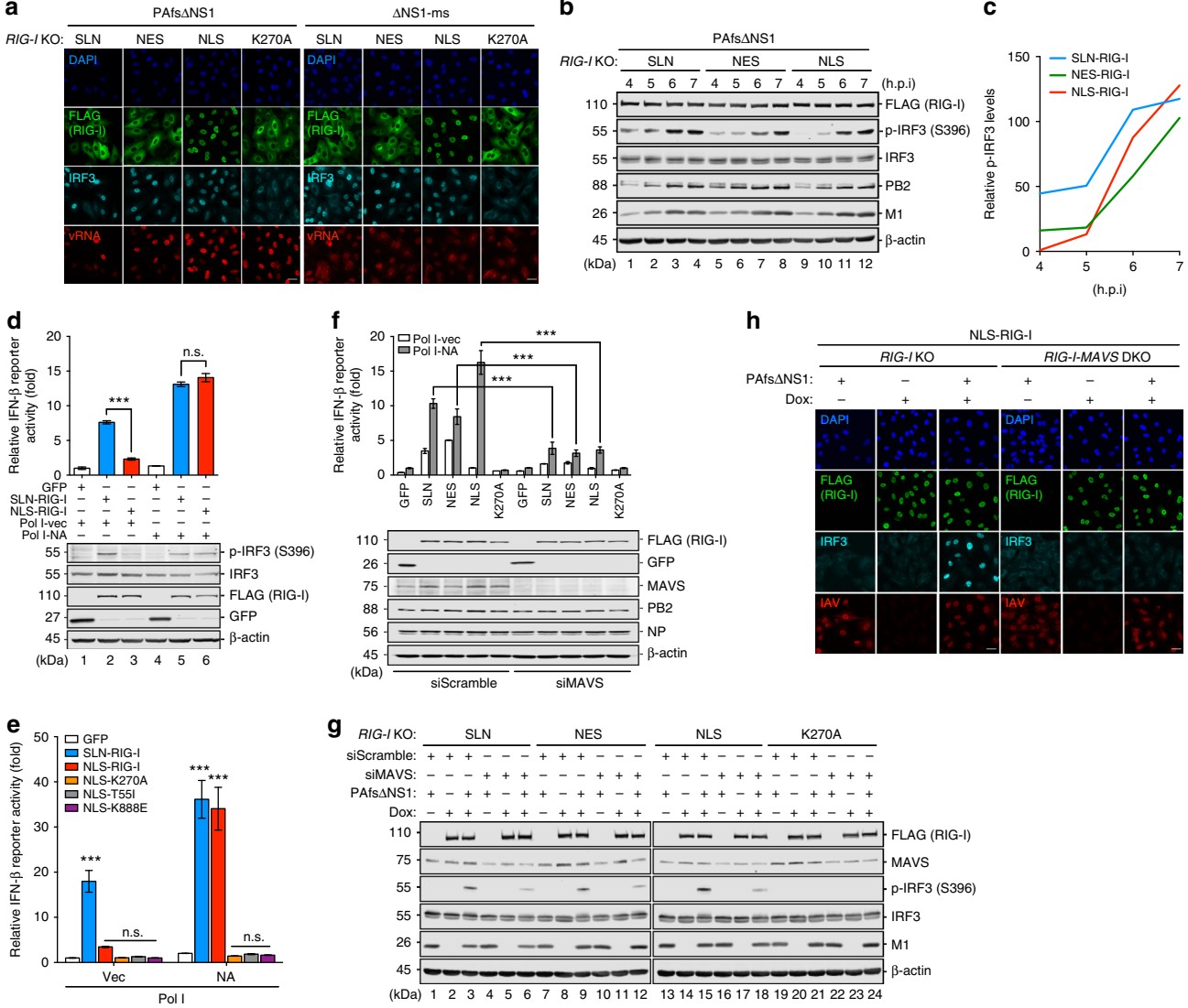

**Fig. 4** Nuclear RIG-I senses IAV via the canonical signaling axis. **a** A549 *RIG-I* KO cell lines inducibly expressing SLN-RIG-I, NES-RIG-I, NLS-RIG-I, or K270A-RIG-I were induced with 1 μg/mL Dox for 4 h followed by infection with PAfsΔNS1 or ΔNS1-ms (MOI = 10) for 7 h. The cells were subjected to immunofluorescence for FLAG-RIG-I (green) and IRF3 (cyan), and FISH for M vRNA (red). **b** Dox-induced A549 cell lines were infected with PAfsΔNS1 virus (MOI = 10). At indicated h.p.i, the protein expression was determined by immunoblotting. **c** Kinetics of IRF3 phosphorylation (**b**) normalized to total IRF3 and β-actin levels were determined by densitometric analysis (ImageJ). **d**, **e** 293T cells were RNP reconstituted with Pol I vector or PR8 NA segment in the presence of GFP, SLN-RIG-I, or NLS-RIG-I (**d**), or in the presence of NLS-RIG-I harboring K270A, T55I, and K888E mutations (**e**) for 24 h. The protein expression was determined by immunoblotting (**d**). RLUs are expressed as fold change relative to the Pol I vector reconstitution in the presence of GFP (**d**, **e**). **f** 293T cells were transfected with 10 nM scramble siRNA (siScramble) or MAVS siRNA (siMAVS) for 24 h followed by RNP reconstitution in the presence of GFP or various RIG-I constructs for another 24 h. RLUs are expressed as that in **d** and **e**. Data are shown as mean ± SD of three experiments. Significant differences were determined by an unpaired Student's *t*-test (**d**) or one-way ANOVA followed by Tukey post-test (**e**, **f**). ***$p < 0.001$; n.s. not significant. **g** Inducible A549 cell lines were transfected with 10 nM indicated siRNA for 40 h, followed by Dox induction (4 h) and PAfsΔNS1 infection (MOI = 10, 6 h). The protein expression was determined by immunoblotting. **h** A549 *RIG-I* KO and *RIG-I-MAVS* DKO cell lines complemented with NLS-RIG-I were left non-induced, or Dox-induced followed by mock or PAfsΔNS1 infection (MOI = 10, 8 h). Immunofluorescence was performed to detect FLAG-RIG-I (green), IRF3 (cyan), and IAV (red). Nuclei were stained with DAPI (blue). Scale bar = 25 μm (**a**, **h**)

and NES-RIG-I at 24 h.p.i (Fig. 5c, lane 10 vs. lanes 2 and 6, compared to non-Dox induced lanes). NLS-RIG-I expression also led to reduced virus titer, albeit to a lesser extent compared to SLN-RIG-I and NES-RIG-I (Fig. 5d). No inhibitory effect on viral protein accumulation and titer was observed in *RIG-I-MAVS* DKO cells expressing either RIG-I (Fig. 5c, d). In line with previous studies[26,45], the signaling-inactive K270A mutant of RIG-I, albeit deficient in IFN-inducing capacity (Fig. 5a, b), consistently exhibited an antiviral effect to some extent. Such signaling-independent activity also required MAVS (Fig. 5c, d).

Collectively, these findings demonstrated that the nuclear-resident RIG-I was able to restrict IAV replication by inducing a MAVS-dependent IFN response and antiviral immunity.

**Nuclear RIG-I shows compartment-specific signaling capacity.** We last assessed whether nuclear RIG-I confers signaling specificity toward viral agonists in the nucleus. To this end, we examined the ability of NLS-RIG-I to sense the cytoplasmic-replicating SeV. As with the conditions for IAV infection

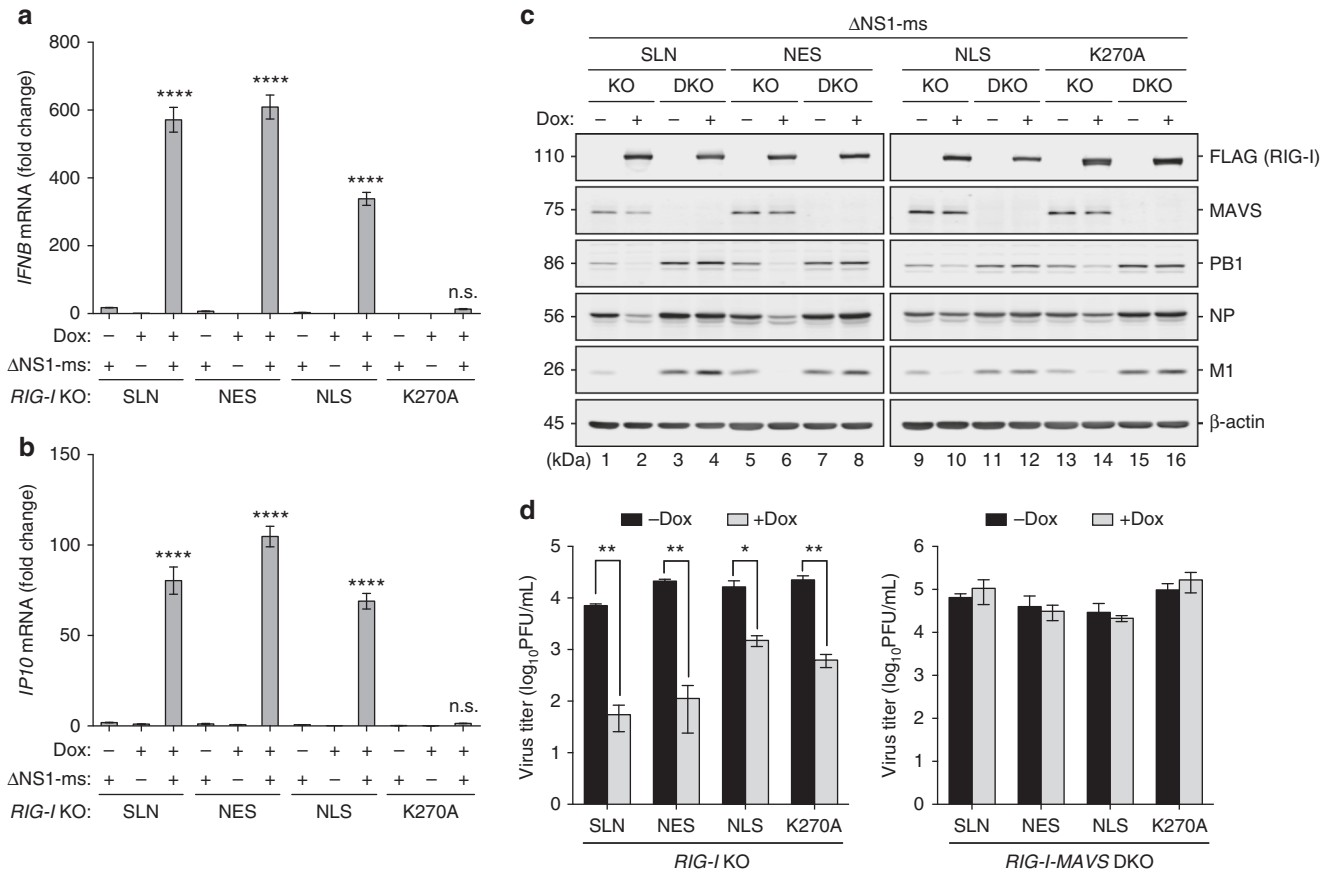

**Fig. 5** Nuclear RIG-I restricts IAV infection. **a**, **b** A549 *RIG-I* KO cell lines inducibly expressing SLN-RIG-I, NES-RIG-I, NLS-RIG-I, or K270A-RIG-I were induced with 1 μg/mL Dox for 4 h, followed by infection with ΔNS1-ms (MOI = 1) for 14 h. The mRNA levels of IFNβ and IP10 (CXCL10) were determined by qRT-PCR. Relative mRNA expression was normalized to GAPDH mRNA levels and expressed using the ΔΔCt method relative to the SLN (with Dox but without infection) condition. **c** Inducible A549 *RIG-I* KO or *RIG-I-MAVS* DKO cell lines were induced as in **a** and **b**, and infected with ΔNS1-ms (MOI = 0.1) for 24 h. Expression levels of FLAG-RIG-I, MAVS, PB1, NP, and M1 were monitored by immunoblotting. **d** Virus titers in the culture supernatants from **c** were determined by plaque assay in MDCK-NS1 cells. Data are mean ± SD of three independent experiments. Significant differences were determined by one-way ANOVA followed by Tukey post-test (**a**, **b**) or an unpaired Student's *t*-test (**d**). *$p < 0.05$, **$p < 0.01$, ****$p < 0.0001$; n.s. not significant

(Fig. 4a), *RIG-I* KO A549 cell lines complemented with the compartment-specific or signaling-inactive RIG-I were infected with SeV and probed for IRF3 nuclear translocation. At 8 h.p.i, while SeV stimulated IRF3 activation in SLN-RIG-I and NES-RIG-I-expressing cells, cells expressing NLS-RIG-I did not respond to SeV replication, resembling that in K270A-RIG-I-expressing cells (Fig. 6a). This null responsiveness by NLS-RIG-I to SeV was further confirmed by quantifying the percentage of cells with the IRF3 nuclear translocation at the single-cell level (Fig. 6b), and IRF3 phosphorylation across the infected cell population (Fig. 6c). At physiologically comparable levels, NLS-RIG-I also failed to sense SeV replication, whereas it efficiently induced antiviral gene expression in response to IAV infection (Supplementary Fig. 7d–g). Together with the efficient IRF3 activation upon IAV replication (Fig. 4a), these results demonstrated a compartment-specific functional sensing by nuclear RIG-I of nuclear-replicating, but not cytoplasmic-replicating, virus.

To further extend the compartment-specific role of nuclear RIG-I to the sensing of a nuclear-replicating DNA virus, we chose HBV since its pgRNA has been characterized as a RIG-I agonist[46]. During the HBV life cycle, pgRNA was transcribed from the viral covalently closed circular DNA in the nucleus followed by nuclear export for reverse transcription[47]. To explore the potential association of nuclear-resident RIG-I with pgRNA, stable Huh-7 cells harboring an HBV genome were subjected to cellular

fractionation and the nuclear extract was immunoprecipitated with the RIG-I antibody. As observed in IAV infection, endogenous RIG-I remained at basal levels in HBV replicon cells (Fig. 6d and Supplementary Fig. 10a); however, the trace of nuclear-resident RIG-I efficiently associated with pgRNA as compared with the IgG control (Fig. 6e). Rather than type I IFN induction, HBV pgRNA sensing by RIG-I stimulated a type III IFN response[46]. In Huh-7.5 cells whose genome encodes a signaling-incompetent endogenous RIG-I[48], ectopic expression of a WT RIG-I mediated efficient IFNλ1 mRNA production in response to transfection with the WT, but not the HBx-deficient HBV genome (Fig. 6f, left). This differential IFNλ1 induction correlated with the relative levels of pgRNA accumulated in transfected cells; the HBx-deficient HBV genome produced significantly less pgRNA (Fig. 6f, right), consistent with a previous report[49]. We next assessed whether nuclear RIG-I activation by HBV pgRNA contributes to type III IFN induction. SLN-RIG-I and NES-RIG-I responded efficiently to pgRNA accumulation leading to IFNλ1 promoter activation from 24 to 48 h.p.t (Fig. 6g) and ISRE promoter activation at 48 h.p.t in 293T cells (Supplementary Fig. 10b). In comparison, NLS-RIG-I exhibited minimal autonomous signaling activity but mediated noticeable IFNλ1 and ISRE promoter activation in 293T cells transfected with the HBV genome (Fig. 6g and Supplementary Fig. 10b). Moreover, NLS-RIG-I sensing of pgRNA required its RNA binding activity, as RIG-I with K888E mutation[43,44] was diminished in IFNλ1 and

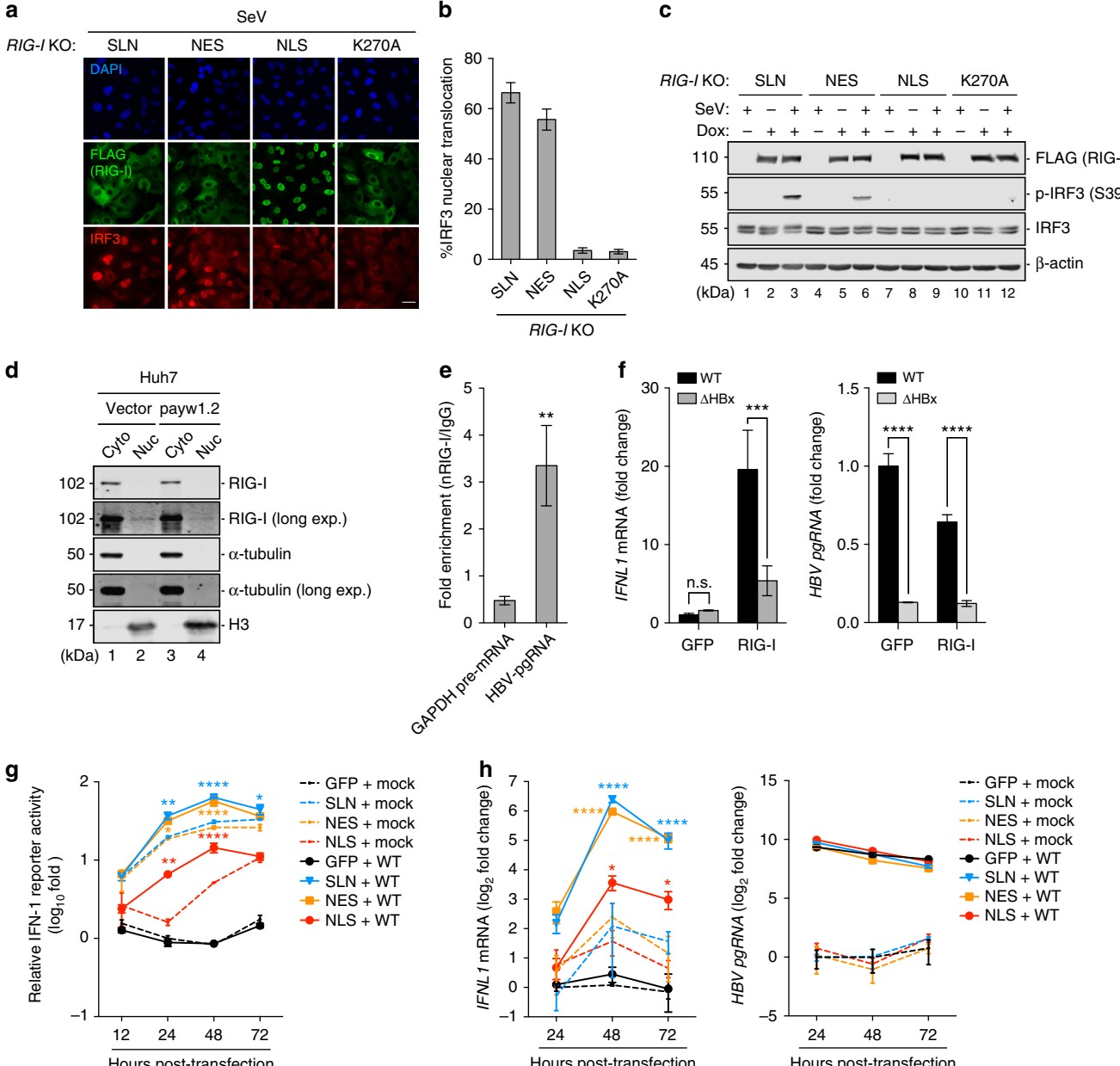

**Fig. 6** Nuclear RIG-I exhibits compartment-specific signaling capacity. **a** A549 *RIG-I* KO cell lines inducibly expressing SLN-RIG-I, NES-RIG-I, NLS-RIG-I, or K270A-RIG-I were induced with 1 μg/mL Dox for 4 h, followed by infection with SeV (50 HAU/mL) for 7 h. Immunofluorescence was performed for FLAG-RIG-I (green) and IRF3 (red). Nuclei were stained with DAPI (blue). Scale bar = 25 μm. **b** The percentage of cells positive for IRF3 nuclear translocation for each cell line (**a**) was calculated by quantifying ten random fields containing ~300 RIG-I-positive cells. **c** Dox-induced A549 cell lines were infected with SeV as in **a**. The protein expression was determined by immunoblotting. **d** The presence of RIG-I in both the cytoplasmic (Cyto) and nuclear (Nuc) fractions of stable Huh-7 cell lines was determined by immunoblotting. **e** RIP was performed using the nuclear extracts of Huh-7-pawy1.2 cells. The levels of HBV pgRNA and GAPDH pre-mRNA in IgG or RIG-I immunoprecipitates were determined by qRT-PCR and are expressed as fold enrichment in the nuclear RIG-I immunoprecipitate relative to IgG. **f** Huh-7.5 cells were cotransfected with WT or ΔHBx HBV genome along with plasmids encoding GFP or RIG-I for 48 h. The levels of IFNλ1 mRNA and HBV pgRNA were determined by qRT-PCR. Fold change is expressed relative to the WT HBV transfection in the presence of GFP. **g** 293T cells were cotransfected with WT HBV genome along with plasmids encoding GFP or various RIG-I constructs, pGL-IFNλ1-fLuc, and pTK-rLuc. RLUs at indicated h.p.t are expressed as fold change relative to the mock transfection in the presence of GFP at 12 h. **h** Huh-7.5 cells were mock transfected or transfected with WT HBV genome in the presence of GFP or various RIG-I constructs. Kinetics of IFNλ1 mRNA and HBV pgRNA expression were determined by qRT-PCR. Fold change is expressed relative to the mock transfection in the presence of GFP at 24 h. Data are mean ± SD of two independent experiments and significant differences were determined by an unpaired Student's *t*-test (**e–h**). *$p < 0.05$, **$p < 0.01$, ***$p < 0.001$, ****$p < 0.0001$; n.s. not significant

ISRE promoter activation (Supplementary Fig. 10b, c). Consistent with that in 293T cells, NLS-RIG-I mediated IFNλ1 mRNA production from 48 to 72 h.p.t in HBV-transfected Huh-7.5 cells, albeit to a lesser extent than SLN-RIG-I and NES-RIG-I (Fig. 6h).

Collectively, although the nuclear RIG-I was less potent than its cytoplasmic counterpart in pgRNA sensing and type III interferon induction, it was appreciably involved in sensing viral agonists derived from within the nucleus.

## Discussion

Our findings on nuclear-resident RIG-I provide advanced insight into the spatiotemporal detection of IAV by RIG-I, which has long been a puzzle given the nuclear replicating nature of IAV. To date, the time frame of RIG-I activation in the course of IAV infection has been attributed to at least three stages. These include RIG-I activation (1) by incoming vRNPs prior to nuclear import[25,26]; (2) during the course of viral RNA synthesis in the nucleus[29]; and (3) by progeny vRNPs following their nuclear export[21]. Among those, controversies remain regarding the relative contribution of the incoming vRNPs and the course of nuclear viral RNA synthesis to RIG-I activation. Although the incoming vRNPs immediately associate with RIG-I[25,26], whether it leads to RIG-I activation and IFN induction remains unclear. In line with recent studies[29,50], our data support the major contribution of nuclear viral RNA synthesis to RIG-I activation. Since RIG-I has been regarded as an exclusive cytoplasmic sensor, previous studies rationalized that the nuclear export of viral agonists is a prerequisite for RIG-I activation. Accordingly, this notion was supported by the observation that a chemical inhibitor causing nuclear retention of viral mRNA abolished IRF3 activation[29]. However, while the nuclear retention of viral mRNA does not necessarily indicate that of other viral RNA species, treatment with the chemical inhibitor substantially diminished viral replication during which the essential agonists activating RIG-I are produced[21,24]. Given the identification of nuclear RIG-I in our study, we propose a shifting paradigm in which the RNA nuclear export is dispensable for RIG-I activation. While the nuclear export of viral agonists is required for the activation of cytoplasmic RIG-I, the nuclear RIG-I could directly sense the nucleus-derived viral agonists and initiate the onset of IFN induction. This additional layer of non-self surveillance in the nuclear compartment is particularly relevant for viruses that have a nuclear phase in their life cycles.

During the course of IAV infection, the relative contributions of the two RIG-I pools are distinct, which fit in the time frame of the nucleocytoplasmic trafficking of vRNP. Nuclear RIG-I accounts for the early association with vRNP while both RIG-I pools are involved in vRNP interaction with the progression of vRNP nuclear export. Although vRNP association indicates closer proximity to the immunostimulatory RNA species such as the genomic vRNA, the early capture of vRNP by nuclear RIG-I did not facilitate a more rapid signaling activation. Either pool of RIG-I by itself mediated a delayed activation of IRF3 during a single-cycle infection, suggesting a coordinated contribution by both the cytoplasmic and nuclear RIG-I to IAV sensing. Prior to the nuclear export of progeny vRNPs, while the cytoplasmic RIG-I provided limited sensing of incoming vRNP and/or undefined viral RNA species exported from the nucleus during nuclear viral replication, the nuclear RIG-I maximized the sensing by 6 h.p.i where nuclear viral RNA synthesis occurred at maximum capacity. During HBV replication, the nuclear RIG-I associated with pgRNA in the nucleus but only modestly induced type III interferons compared with the cytoplasmic RIG-I. As the nuclear RNA synthesis and export mechanisms of HBV differ substantially from that of IAV, the exact mode of nuclear RIG-I activation by HBV merits further investigation.

Currently, no clear nucleocytoplasmic shuttling activity is defined for RIG-I. While a previous study proposed a RIG-I nuclear translocation model driven by its interaction with IAV vRNP components[51], we demonstrated nuclear localization of RIG-I even in the absence of viral infection. Mechanistically, nuclear RIG-I activation appears to require its shuttling out of the nucleus, as all its downstream adapters, including TRIM25 and MAVS, are located in the cytoplasm. However, this translocation may not be absolutely required if a complete signaling cascade

could be established within the nucleus. Interestingly, the recently identified nuclear TRIM25 fits into our finding, which would facilitate nuclear RIG-I activation[52]. Upon IAV infection, no noticeable change in MAVS localization was observed, suggesting there is no reorganization of MAVS to facilitate interaction with activated nuclear RIG-I. Previous studies reported that MAVS filaments exhibited intertwined network with perinuclear distribution, indicating a close association with nuclear membrane[53,54]. Such intimate contact and even fusion of mitochondria with nuclear membrane have long been observed to provide an essential gateway for energy delivery dictating cellular mRNA transport[55]. Thus, it is conceivable that the activated nuclear RIG-I interacts with the mitochondria and downstream adapters at the nuclear membrane interface at regions of greater permeability. Interestingly, IAV infection induces Nup153 nucleoporin degradation resulting in nuclear pore enlargement to passively facilitate vRNP export[56]. Therefore, we propose that during IAV infection, an altered nuclear membrane architecture may facilitate the formation of a sensing milieu for nuclear RIG-I at the perinuclear region (Supplementary Fig. 11). Such perinuclear membrane-associated pattern also emerges for cytoplasmic RIG-I during acute SeV infection[53,57]. Moreover, the requirement of compromised integrity of nuclear envelope has also been shown for herpesvirus-induced relocalization of host 5S ribosomal RNA pseudogene transcripts activating RIG-I[58].

Overall, the identification and characterization of nuclear-resident RIG-I in relation to IAV infection provide the first evidence of a non-self RNA sensing paradigm in the nucleus. Compared to other members of the DExD/H-box RNA helicase family which are mostly nuclear shuttle or exclusively nuclear proteins[59], the nuclear localization of RIG-I likely implies a remnant function retained in the nucleus during its evolution toward the cytoplasmic compartment. Here we define such function as the active sensing of nucleus-derived viral RNA contributing to antiviral immunity. Understanding the nuclear roles for other RLRs and RNA sensors would facilitate the establishment of a complete nuclear RNA sensing paradigm and the coordination of RNA sensing pathways from different cellular compartments culminating in virus restriction.

## Methods

**Cell culture**. Madin-Darby canine kidney (MDCK, ATCC, #CRL-2936) and human embryonic kidney (HEK293T, ATCC, #CRL-3216) cells were maintained in minimal essential medium Eagle (MEM, Sigma) and Dulbecco's Modified Eagle's Medium (DMEM, Sigma) supplemented with 10% FBS and 50 μg/mL gentamicin, respectively. The MDCK cell line inducibly expressing the NS1 protein of A/Puerto Rico/8/34 (PR8) was constructed by lentiviral transduction of MDCK cells with a Tet-On inducible lentiviral vector containing the PR8 NS1 open reading frame (ORF). The MDCK-NS1 cell line was cultured in MEM supplemented with 10% tetracycline-free FBS (Clontech), 50 μg/mL gentamicin, and 5 μg/mL puromycin (Invivogen). Human lung epithelial (A549, ATCC, #CRM-CCL-185) cells and its derivative knock-out (KO) cells were maintained in Ham's F-12K (Kaighn's) medium (Gibco) containing 10% FBS and 50 μg/mL gentamicin. A549 *RIG-I* KO and *RIG-I-MAVS* DKO cells inducibly expressing scrambled tagged (SLN)-RIG-I, NES-RIG-I, NLS-RIG-I, and RIG-I with K270A mutation were cultured in F-12K with 10% tetracycline-free FBS, gentamicin, and 2 μg/mL puromycin. Human hepatoma Huh-7.5 cells and Huh-7 cells stably transfected with an empty vector or a greater-than-unit-length HBV genome (pawy1.2) carrying a luciferase reporter were maintained in DMEM supplemented with 10% FBS, with the stable cell lines additionally supplemented with 100 μg/mL hygromycin B (Invivogen). All cell cultures were maintained at 37 °C in a humidified 5% $CO_2$ atmosphere and were monthly tested for mycoplasma contamination.

**DNA constructs and transfection**. The human RIG-I ORF was amplified from the pEF-flagRIG-I[41] and subcloned into pCMV-3xFLAG. To generate compartment-specific RIG-I constructs, a monopartite SV40 large T antigen NLS (PKKKRKV) and a nuclear factor erythroid 2-related factor 2 NES (LKKQLSTLYL) were inserted between the 3×FLAG tag and RIG-I ORF, respectively, giving rise to pCMV-3×FLAG-NLS-RIG-I and pCMV-3×FLAG-NES-RIG-I. In parallel, an inverted NLS sequence (NLPLFLR) was inserted at the same location and designated SLN-RIG-I as a scrambled control. Site-directed mutagenesis on

pCMV-3×FLAG-NLS-RIG-I (T55I, K270A, and K888E) or on viral polymerases in the backbone of pcDNA3.1 (PB2-R142A, PB2-E361A, PB1-D445A/D446A, PA-D108A, and PA-E410A) was introduced by overlapping PCR. Episomal constructs, with Tet-On inducible expression of 3×FLAG-SLN-RIG-I, NES-RIG-I, NLS-RIG-I, or K270A-RIG-I, were constructed by cloning the respective ORFs downstream of a minimal CMV promoter that is behind a tetracycline responsive element (TRE promoter). This vector also contains expression cassettes for the transactivator rtTA, puromycin resistance, and the Epstein-Barr virus nuclear antigen 1 (EBNA1). For *RIG-I* KO, an annealed oligonucleotide pair encoding the 20-nt guide sequences (GGGTCTTCCGGATATAATCC) targeting the exon 1 of human RIG-I was ligated in between the two *BbsI* sites of the CRISPR/Cas9 plasmid pSpCas9–2A-Puro (PX459) v2.0 (Addgene #62988). The correct sequence of all constructs was confirmed by DNA sequencing (Eurofins). The sequences of cloning primers are listed in Supplementary Table 1. Transient transfection of 293T and Huh-7.5 cells was performed using TransIT-LT1 (Mirus Bio) and jetPEI (Polyplus) as per manufacturer's instructions, respectively.

**Viruses and reverse genetics**. The recombinant viruses were generated by the eight-plasmid reverse-genetics system as previously described with slight modification[60]. The set of pHW plasmids (pHW191-198) encoding the eight A/Puerto Rico/8/34 (H1N1) segments was used as the backbone for mutant virus generation. PA-X/NS1 double-deficient PR8 (PAfsΔNS1) contains PA segment with abolished +1 ribosomal frameshifting for X-ORF translation[34], and NS segment with the intron region deleted. Rescue of PA-X/NS1 double-deficient virus was performed in co-cultured MDCK-NS1/293T cells in Opti-MEM (Gibco) containing 0.2% bovine serum albumin (BSA), 1 μg/mL L-1-tosylamide-2-phenylethyl chloromethyl ketone (TPCK)-trypsin, and 1 μg/mL Dox (Sigma). The rescued virus was propagated and titrated in MDCK-NS1 cells. The genome authenticity of segments 3 and 8 of the mutant virus was verified by RT-PCR followed by DNA sequencing. The widely used PR8ΔNS1 virus (ΔNS1-ms)[33] was propagated in MDCK-NS1 cells. A/Victoria/3/75 (H3N2) and a low pathogenic A/chicken/British Columbia/CN-6/2004 (H7N3) were propagated in MDCK cells and embryonated chicken eggs, respectively. SeV (Cantell strain) was obtained from ATCC (VR-907) and propagated in embryonated chicken eggs. All virus infections were performed at the indicated multiplicity of infection (MOI) by incubating the virus inoculum diluted in Dulbecco's Phosphate-Buffered Saline containing calcium and magnesium (DPBS +/+, Hyclone) with cells for 1 h. The inoculum was then removed and replaced with plain medium containing 0.2% BSA and 1 μg/mL TPCK-trypsin (or 0.25 μg/mL for A549 cells and its derivatives). TPCK-trypsin was omitted from the culture medium for single-cycle infections.

**A549 cells inducibly expressing compartment-specific RIG-I**. The CRISPR-Cas9 system was employed to generate *RIG-I* KO, *MAVS* KO, and *RIG-I-MAVS* DKO A549 cells[61]. Briefly, A549 cells were transfected with pSpCas9-2A-Puro (PX459) v2.0 plasmid harboring the sgRNA sequence using lipofectamine 3000 transfection reagent (Invitrogen) as per manufacturer's instructions. At 24 h post-transfection (h.p.t.), cells were subjected to puromycin selection at a concentration of 2 μg/mL. Two days later (72 h.p.t.), puromycin was removed from the culture medium to avoid genome integration of the CRISPR plasmid. After 2 weeks, monoclonal cell lines were isolated by limiting dilution at a density of 0.5 cells/well followed by clonal expansion for 3 weeks. Fourteen selected clones were characterized by examining RIG-I expression level induced by SeV infection (100 HAU/mL) and IFNβ1a treatment (500 U/mL, IF014, Millipore). Genomic DNA was further extracted from two of the selected clones using a DNeasy Blood & Tissue Kit (Qiagen) and amplified for the region flanking RIG-I exon 1. One monoclonal cell line (KO3) exhibited a homozygous single-nucleotide insertion within the sgRNA targeting site. To generate *MAVS* KO and *RIG-I-MAVS* DKO A549 cells, WT A549 or the KO3 cells were transfected with plasmids harboring three sgRNA sequences targeting human MAVS (Santa Cruz). Monoclonal cell lines were selected by limiting dilution and characterized by the ablation of MAVS expression. The monoclonal *RIG-I* KO (KO3) and *RIG-I-MAVS* DKO (DKO4) cell lines were further used to construct reconstituted A549 cell lines inducibly expressing compartment-specific RIG-I. To avoid any off-target effect conferred by lentiviral vector-based delivery of exogenous gene expression, we chose a non-integrating episomal expression approach to construct these cell lines. The KO3 and DKO4 cells were transfected with episomal plasmids expressing FLAG-tagged SLN-RIG-I, NES-RIG-I, NLS-RIG-I, or K270A-RIG-I under the control of a TRE promoter followed by selection with 2 μg/mL puromycin for 4 weeks. After the establishment of persistent episomal replication, the stable cell lines were characterized by examining FLAG-RIG-I expression and localization in the presence or absence of 1 μg/mL Dox.

**Co-immunoprecipitation and immunoblotting**. A549 or HEK293T cells were infected with indicated viruses or RNP reconstituted followed by crosslinking in PBS (−/−) containing 1 mM dithiobis[succinimidyl propionate] (DSP, Sigma) for 30 min on ice. Cells were subjected to cellular fractionation as described elsewhere[52] and the nuclear pellets were extracted at 4 °C for 30 min with a buffer containing 50 mM Tris-HCl (pH 7.4), 200 mM NaCl, 1% NP-40, 1 mM EDTA, 50 μg/mL RNase A, and 1× cOmplete protease inhibitors (Roche). The nuclear

extracts were further pre-cleared with 20 μL protein G Dynabeads (Invitrogen) at 4 °C for 30 min. For co-immunoprecipitation, 35 μL protein G Dynabeads were conjugated with 3 μg of either mouse IgG1 isotype control (CST) or mouse anti-RIG-I antibody (1C3, Millipore) at 4 °C for 1 h. Pre-cleared lysates were then incubated with the conjugated beads at 4 °C for 16 h. On the next day, the beads were washed four times with the wash buffer (50 mM Tris-HCl, pH 7.4, 150 mM NaCl, 1% NP-40, 1 mM EDTA) and eluted in 1× Laemmli buffer containing 355 mM β-mercaptoethanol. Immunoblotting was performed as previously described[24]. Briefly, protein samples were resolved by sodium dodecyl sulfate polyacrylamide gel electrophoresis (SDS-PAGE), transferred onto nitrocellulose membranes, and probed with indicated primary antibodies including rabbit anti-RIG-I (EPR18629, Abcam, 1:500), mouse anti-FLAG (M2, Sigma, 1:1000), rabbit anti-FLAG (CST, 1:1000), rabbit anti-Phospho-IRF3 (Ser396) (D6O1M, CST, 1:1000), rabbit anti-IRF3 (D6I4C, CST, 1:1000), mouse anti-GFP (4B10, CST, 1:1000), mouse anti-α-tubulin (Abcam, 1:1000), mouse anti-β-actin (8H10D10, CST, 1:1000), rabbit anti-Lamin B1 (ab16048, Abcam, 1:1000), and rabbit anti-histone H3 (D1H2, CST, 1:1000). Rabbit polyclonal antisera against IAV PB2, PA, NP, and M1 were raised in-house, and rabbit polyclonal antibody against IAV NS1 (PA5-32243, 1:500) was obtained from Pierce. For cellular fractionation, α-tubulin and histone H3 were used as markers for the cytoplasmic and nuclear fractions, respectively. Following incubation with donkey anti-rabbit IRDye 680RD and donkey anti-mouse IRDye 800CW secondary antibodies (LI-COR), membranes were visualized with an Odyssey infrared imaging system (LI-COR). All uncropped immunoblots are presented in Supplementary Fig. 12.

**RNA immunoprecipitation (RIP)**. For RIP, Huh-7-HBV-payw1.2-Luc cells were DSP-crosslinked and an aliquot was saved for input RNA extraction. The remaining cells were subjected to cellular fractionation. The nuclear pellets were extracted with a buffer containing 50 mM Tris-HCl (pH 7.4), 200 mM NaCl, 1% NP-40, 1 mM EDTA, 100 μg/mL *E. coli* tRNA (Roche), 100 U/mL RNasin (Promega), and 1× EDTA-free protease inhibitors at 4 °C for 1 h. The nuclear extracts were pre-cleared with protein G Dynabeads for 30 min and incubated with IgG1 or RIG-I antibody conjugated Dynabeads overnight at 4 °C. The beads were extensively washed with the wash buffer containing 50 mM Tris-HCl (pH 7.4), 150 mM NaCl, 0.5% NP-40, 1 mM EDTA, 100 μg/mL tRNA, and 40 U/mL RNasin, followed by proteinase K digestion in a buffer containing 20 mM Tris-HCl (pH 7.4), 100 mM NaCl, 2 mg/mL proteinase K (NEB), 1% NP-40, 0.1% SDS, 100 U/mL RNasin, 100 μg/mL tRNA, and 1 mM EDTA at 37 °C for 1.5 h. The immunoprecipitated and input RNA were extracted using the TRIzol Reagent (Ambion) as per manufacturer's instructions and reverse transcribed using oligo(dT) and random primers with SuperScript IV Reverse Transcriptase (Invitrogen). The levels of HBV pgRNA in IgG1 and RIG-I immunoprecipitates were determined by qRT-PCR using specific primers targeting the ε regions of pgRNA as previously described[46]. The levels of GAPDH pre-mRNA in both immunoprecipitates were determined in parallel as a negative control. The amount of each immunoprecipitated RNA was normalized to that in the input RNA, and was expressed as fold enrichment in the nuclear RIG-I immunoprecipitate relative to the IgG1 control.

**Immunofluorescence and RNA FISH**. A549 or HEK293T cells were grown on poly-L-lysine-coated 8-well or 16-well chamber slides (Nunc), and infected or transfected as indicated. At indicated time points post-infection or post-transfection, cells were rinsed twice with DPBS (+/+) and fixed with 4% formaldehyde (methanol-free) in DPBS (+/+) for 10 min at room temperature (RT). The cells were subsequently permeabilized with 0.2% Triton X-100 for 5 min and blocked with 5% BSA in DPBS (+/+) at RT for 1 h. Following incubation with primary antibodies including goat anti-IAV (AB1074, Millipore, 1:2000), mouse anti-IAV NP (AA5H, AbD Serotec, 1:500), rabbit anti-IRF3 (D6I4C, CST, 1:400), mouse anti-RIG-I (1C3, Millipore, 1:250), mouse anti-FLAG (M2, Sigma, 1:500), mouse anti-MAVS (E-3, Santa Cruz, 1:200), goat anti-FLAG (ab1257, Abcam, 1:500), and rabbit anti-β-tubulin (9F3, CST, 1:100), the cells were washed and incubated with secondary antibodies diluted in 5% BSA for 1 h. Secondary antibodies used include donkey anti-mouse Alexa Fluor 488, donkey anti-rabbit Alexa Fluor 546, donkey anti-rabbit Alexa Fluor 594, and donkey anti-goat Alexa Fluor 633 (Invitrogen, 1:500). After nuclear staining with 4′,6-diamidino-2-phenylindole (DAPI) for 5 min, the cells were washed four times with DPBS (+/+), once with RNase-free water, and mounted in ProLong Gold antifade mountant (Invitrogen). The FISH analysis was performed as previously described with modifications[62,63]. Briefly, following secondary antibody staining, the cells were subjected to another fixation step with 4% formaldehyde in DPBS (+/+) for 10 min, washed once, and equilibrated with 2 × SSC (300 mM sodium chloride, 30 mM sodium citrate) with 10% formamide for 10 min. To detect IAV vRNA in infected or RNP-reconstituted cells, a pool of 39 probes against viral M segment was custom designed (Stellaris, Biosearch Technologies) and the cells were incubated with 125 nM Quasar 670-conjugated probes in a hybridization buffer (10% dextran sulfate, 2 mM vanadyl–ribonucleoside complex (VRC), 0.02% RNA-free BSA, 1 mg/mL *E. coli* tRNA, 2 × SSC, and 10% formamide) in a humidified chamber at 28 °C for 16 h. On the next day, the cells were washed with 2 × SSC containing 10% formamide and 2 mM VRC, stained with DAPI, and mounted as described above. Confocal microscopy was carried out using a Leica TCS SP8 confocal microscope, equipped with a Diode (405 nm), an Argon (458, 476, 488, 496, 514 nm), a DPSS (561 nm),

and a Helium/Neon (633 nm) lasers. Image processing, quantification, and colocalization analysis were performed using the Fiji version of ImageJ (NIH).

**RNP reconstitution and luciferase reporter assay.** The RNP reconstitution was performed as previously described[24]. To monitor viral polymerase activity in the presence of compartment-specific RIG-I, HEK293T cells were transfected with the set of plasmids encoding the vRNP components (pcDNA-PB2, pcDNA-PB1, pcDNA-PA, pcDNA-NP, and pHH21-NP-fLuc), a constitutively expressing *Renilla* luciferase reporter plasmid (pTK-rLuc), alongside with plasmids expressing various RIG-I (pCMV-3×FLAG-SLN/NES/NLS-RIG-I) or GFP. To measure IFNβ promoter activation upon RNP reconstitution, 293T cells were transfected with plasmids as above except that the pHH21-NP-fLuc was replaced by pHH21 plasmids encoding each of the authentic viral RNA (pHH21-PB2 to NS), and a human IFNβ promoter-driven luciferase reporter plasmid (p125Luc) was supplemented. Transcription-defective (PB2-E361A, PA-D108A) or replication-defective (PB2-R142A, PA-E410A) polymerase constructs were also included in place of the WT polymerase to determine the requirement of viral transcription and replication for IFNβ promoter activation upon RNP reconstitution. The relative luciferase activity (RLU) was determined at 24 h.p.t using the Dual-Luciferase Reporter Assay System (Promega). To test the signaling competency of N-terminally tagged RIG-I constructs, 293T cells were transfected with various RIG-I plasmids or GFP together with p125Luc and pTK-rLuc for 6 h. The cells were further left untreated or stimulated with a 19-bp 5′ppp-dsRNA (Invivogen) using Lipofectamine RNAiMAX (Invitrogen) for 18 h. For IFNλ1 promoter activation by HBV, 293T cells were transfected with WT HBV (payw1.2) or ΔHBx HBV (payw1.2*7) genome[64] along with pGL4.18-λ1(−1070/+46) and pTK-rLuc in the presence of compartment-specific RIG-I or GFP. The RLU was determined at the indicated time points after transfection.

**Cellular fractionation, strand-specific qRT-PCR, and qRT-PCR.** HEK293T cells were transfected with plasmids encoding vRNP components in conjunction with GFP or indicated RIG-I constructs for 24 h. Reconstitution in the presence of a catalytically inactive PB1 subunit (PB1a, D445A/D446A) was performed in parallel to set the background RNA levels. Reconstituted cells were washed twice in PBS (−/−) and resuspended in a hypotonic buffer containing 10 mM HEPES (pH 7.4), 10 mM KCl, 1.5 mM MgCl₂, 1 mM DTT, 1× EDTA-free protease inhibitors, and 100 U/mL RNasin. After incubation on ice for 10 min, NP40 was added to a final concentration of 0.5% and the cell membrane was disrupted by vortexing at high speed for 15 s. The disrupted cells were immediately spun down at $3500 \times g$ for 10 min and the supernatant was saved as the cytoplasmic fraction. The pellet was washed once more with the hypotonic buffer and saved as the nuclear fraction. Both fractions were resuspended in the TRIzol Reagent and subjected to RNA extraction per manufacturer's instruction. Extracted RNA from both fractions were resuspended in equal volume of water and stored at −80 °C. Strand-specific qRT-PCR was performed as previously described with modification[24]. Briefly, 5% of both cytoplasmic and nuclear RNA fractions were reverse transcribed using tagged strand-specific primers (sequences available upon request) for viral RNA species or oligo(dT) plus random primer for total RNA. Fold change of viral RNA levels was normalized to the cytoplasmic *GAPDH* mRNA levels and expressed using the ΔΔCt method relative to the nuclear RNA fraction of the PB1a reconstitution. Successful cellular fractionation was determined by immunoblotting for Lamin B1 and α-tubulin in the nuclear and cytoplasmic fractions, respectively. Additionally, the exclusive presence of *GAPDH* pre-mRNA in the nuclear, but not the cytoplasmic fraction, served as a control at the RNA level. To determine mRNA levels of *IFNB1* and C-X-C motif chemokine 10 (*CXCL10*; *IP10*) induced by the ΔNS1-ms virus or *IFNL1* induced by HBV replication, total RNA was extracted from infected or transfected cells using an RNeasy Plus Mini Kit (Qiagen) as per manufacturer's instructions. Reverse transcription was carried out using oligo(dT) and SuperScript IV Reverse Transcriptase. qPCR was performed using Power SYBR Green PCR Master Mix (Applied Biosystems) on a StepOnePlus Real-Time PCR System (Applied Biosystems). Relative mRNA expression was normalized to mRNA levels of housekeeping genes (*GAPDH* for IAV infection or *GUSB* for HBV genome transfection) and expressed using the ΔΔCt method relative to the conditions as indicated. The sequences of all qPCR primers are listed in Supplementary Table 2.

**RNA interference.** The Silencer Select siRNA negative control and siRNA against human MAVS (sense: 5′-GGGUUCUUCUGAGAUUGAAtt-3′; antisense: 5′-UUCAAUCUCAGAAGAACCCag-3′) were obtained from Invitrogen. Briefly, HEK293T and inducible A549 cells were transfected with 10 nM siRNAs using Lipofectamine RNAiMAX (Invitrogen) and TransIT-X2 reagents (Mirus Bio), respectively. At 24 or 40 h post-transfection, cells were reconstituted with vRNP or infected with the mutant virus as indicated.

**Statistical analysis.** The statistical significance of differences was calculated using GraphPad Prism 7 (GraphPad Software, Inc., USA) with an unpaired Student's *t*-test, one-way or two-way ANOVA followed by the Tukey or Sidak post-hoc test to obtain the *p*-value. Data are shown as mean ± SD of three independent experiments performed in triplicates unless otherwise indicated. Significant differences between groups are denoted by $*p < 0.05$, $**p < 0.01$, $***p < 0.001$, or $****p < 0.0001$.

**Data availability.** All relevant data are available upon request from the corresponding author.

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

## Acknowledgements

We thank S. Pleschka (Justus-Liebig University Giessen) for pMP plasmids harboring A/Victoria/3/75 (H3N2) segments, T. Fujita (Kyoto University) for pEF-flagRIG-I plasmid, F. Zhang (Broad Institute) for pSpCas9(BB)-2A-Puro (PX459) v2.0 plasmid, B. L. Slagle (Baylor College of Medicine) for payw1.2 and payw1.2*7 plasmids, T. Kameyama (Hokkaido University) for pGL4.18-λ1(-1070/+46) plasmid, and A. García-Sastre (Icahn School of Medicine at Mount Sinai) for the ΔNS1-ms virus. G.L. is partially supported by the Vaccinology and Immunotherapeutics (V&I) Scholarship from the School of Public Health, University of Saskatchewan. This work was supported by a NSERC grant to Y.Z.

## Author contributions

G.L. and Y.Z. conceived the project, designed experiments, and wrote the manuscript. G.L. performed all aspects of the experiments unless otherwise indicated. Y.L. performed DI detection and assisted with virus titration, immunofluorescence, and luciferase reporter assays. S.N.T.R. assisted with immunofluorescence in HeLa cells. F.X., Q.W., Z.L., R.B., and Q.L. provided reagents and resources. All authors discussed data and commented on the manuscript.

## Additional information

**Competing interests:** The authors declare no competing interests.

