## [Peer Review File · Nature Communications]

Reviewers' comments:

Reviewer #1 (Remarks to the Author):

This manuscript addresses a question that has generated considerable interest over the years. Interferon induction by influenza virus is a widely-studied area and it is known that RIG-I is the major, if not the only, PRR that senses influenza PAMPs. However, RIG-I is referred to as a cytoplasmic receptor whilst influenza virus transcribes and replicates in the nucleus. There are a number of reports that offer solutions to this dilemma, but this report offers very convincing evidence for the existence of a nuclear pool of RIG-I that responds to influenza infection to activate innate immune responses, and furthermore that this pool of RIG-I is the physiological mediator of anti-influenza virus responses. The experiments are extremely well-designed and conclusive, and the paper is largely well-written. I have two specific comments:

1. It is not clear why RIG-I levels remain at basal values in A549 cells infected by deltaNS1 influenza virus (Figure 2A). Since deltaNS1 virus has lost the RIG-I antagonist the virus should activate RIG-I, cause IRF3 to translocate to the nucleus, and interferon should be produced. Indeed, the demonstration that RIG-I levels increase in neighbouring, uninfected, cells indicates that interferon is indeed being produced. However, since the induction of both interferon and RIG-I in response to viral infection are RIG-I-dependent and IRF3-dependent events, RIG-I should be up-regulated in the deltaNS1-infected cells. The text is confusing at this point since it introduces a second virus that also lacks a polymerase-associated gene. The logic behind the need for this escaped me, so some clarification of this part of the text would be helpful.

2. In lines 199/200 it is argued that early induction of IRF3 by Sendai virus argues against the presence of DI particles. Having worked on Sendai virus for about 35 years I would say it argues strongly for the presence of DI particles! The authors are using the Cantell strain, a strain selected as a good inducer of interferon because it has a propensity to generate DI particles. However, since I cannot see how it matters for these experiments what kind of inducer/PAMP the Sendai virus is making the sentence on DI particles should just be removed.

Reviewer #2 (Remarks to the Author):

Key results: Please summarise what you consider to be the outstanding features of the work. This manuscript reported that nucleus-resident RIG-I senses nonself RNA and activates the MAVS-dependent signaling cascade to induce IFN expression. An elegant experimental design entailing NLS and NES was used to discern the distinct function of nuclear and cytoplasmic RIG-I in sensing Influenza A virus (nuclear) and Sendai virus (cytoplasmic). A novel signaling model was also proposed, which could be significant for related studies in antiviral immunology. These findings are important to advance our understanding in the antiviral activity of RIG-I. However, there are a few major concerns that the author may consider to address.

The amount of nuclear RIG-I, especially when compared to the cytoplasmic RIG-I, is relatively small or minute. Nuclear RIG-I was more apparent at later time points during viral infection, likely due to further induction by IFN. This raises the question whether cytoplasmic RIG-I is sufficient or nuclear RIG-I is necessary for sensing viral RNA, such as those of IAV. Apparently, the authors have supplied a great deal of data to support their conclusions. One major concern is whether the "reconstituted" expression level of nuclear RIG-I is physiologically comparable to the amount of nuclear endogenous RIG-I of virus-infected cells. Such comparison may be crucial to consolidate the authors' conclusion.

Given the concept that nuclear RIG-I signals through the mitochondrion-localized MAVS, it is

conceivable that nuclear RIG-I shuttles into the cytoplasm during IAV infection. This data will be important to further validate the role of a nucleus-localized RIG-I in sensing IAV RNA. The authors rely on IFN reporter assay for the antiviral immune response downstream of RIG-I. When necessary, key experiments have to be validated by real-time PCR analysis of key effectors downstream of the RIG-I signaling pathway.

Validity: Does the manuscript have flaws which should prohibit its publication? If so, please provide details.

There are a few major concerns that the authors need to address, please see main summary.

Originality and significance: If the conclusions are not original, please provide relevant references. On a more subjective note, do you feel that the results presented are of immediate interest to many people in your own discipline, and/or to people from several disciplines?

Yes.

Data & methodology: Please comment on the validity of the approach, quality of the data and quality of presentation. Please note that we expect our reviewers to review all data, including any extended data and supplementary information. Is the reporting of data and methodology sufficiently detailed and transparent to enable reproducing the results?

The experiments are well and cleverly designed. Their results are clear-cut and interpreted properly.

Appropriate use of statistics and treatment of uncertainties: All error bars should be defined in the corresponding figure legends; please comment if that's not the case. Please include in your report a specific comment on the appropriateness of any statistical tests, and the accuracy of the description of any error bars and probability values.

Scientific statistical analysis is well performed, except the correlation results. It would be better to show the correlation coefficient or its equivalent to support the positive/negative correlation.

Conclusions: Do you find that the conclusions and data interpretation are robust, valid and reliable?

Yes, the authors' conclusions are generally supported by their experimental results. However, the function of nuclear RIG-I may have to be better studied with a controlled expression comparable to the endogenous level of nuclear RIG-I.

Suggested improvements: Please list additional experiments or data that could help strengthening the work in a revision.

Please see comments in the summary that protein expression level may be better controlled.

References: Does this manuscript reference previous literature appropriately? If not, what references should be included or excluded?

Yes.

Clarity and context: Is the abstract clear, accessible? Are abstract, introduction and conclusions appropriate?

Introduction: Introduction should be condensed to improve the readability. For example, the first two paragraphs can be combined to one that focuses on the RNA sensing studies.

Line 83: Please use most-studied rather than "best-studied".

Line 95: Please define "vRNP" at the first place.

Results: For SLN- and NES experiments, why use doxycycline as stimuli, not IFN β stimulation or SeV (this is used to induce RIG-I expression)? In line 154-155, the IAV infection (also refer to Figure 1E) is a sequential experiment, need an explanation.

Line 120: ...we subjected A549 cells to SeV infection or IFN β stimulation... to correspond the order of following description.

Line 127: Please define "IRF3".

Figure 2: Please be consistent with the color coatings. Use same color to represent the IRF3 throughout the figure or manuscript (if possible?).

The RIG-I expression in Δ NS1-ms infected condition was not apparent at 8 h.p.i, but it was quite distinct at 14 h.p.i. comparing to mock-infected group in the Supplementary Fig. 3A. Please revise the statement in Line 169-171 and following data interpretation. Because of the life cycle?

Line 201-203: The statement claims the two mutants indeed activate RIG-I, however contradict to the earlier description in Line 166-171.

Line 233-235: Please describe the reconstitution system in the Methods.

Line 253-254: Please also perform the statistical analysis, supporting the NLS-RIG-I inactive. Also, the y-axle in Figure C and D are 20 times difference, how could compare them in a comparable way?

Line 277-280: When authors claim the positive/negative correlation, could authors also provide quantitative results, such as correlation coefficient or similar indicators.

I wonder whether the expression of p-IRF3 and IRF3 in cytoplasmic and nucleus fractions in NES and NLS differ? In particular, to explore the "cooperative IFN induction along with its cytoplasmic counterpart".

Line 360: ...across infected cell population? Please clarify.

Discussion: Please consider remove the first paragraph in the Discussion, which is redundant, repeating the description of abstract and the part of the introduction.

Please indicate any particular part of the manuscript, data, or analyses that you feel is outside the scope of your expertise, or that you were unable to assess fully.

Reviewer #1 (Remarks to the Author):

Comments: *This manuscript addresses a question that has generated considerable interest over the years. Interferon induction by influenza virus is a widely studied area and it is known that RIG-I is the major, if not the only, PRR that senses influenza PAMPs. However, RIG-I is referred to as a cytoplasmic receptor whilst influenza virus transcribes and replicates in the nucleus. There are a number of reports that offer solutions to this dilemma, but this report offers very convincing evidence for the existence of a nuclear pool of RIG-I that responds to influenza infection to activate innate immune responses, and furthermore that this pool of RIG-I is the physiological mediator of anti-influenza virus responses. The experiments are extremely well-designed and conclusive, and the paper is largely well-written. I have two specific comments:*

1. It is not clear why RIG-I levels remain at basal values in A549 cells infected by deltaNS1 influenza virus (Figure 2A). Since deltaNS1 virus has lost the RIG-I antagonist the virus should activate RIG-I, cause IRF3 to translocate to the nucleus, and interferon should be produced. Indeed, the demonstration that RIG-I levels increase in neighboring, uninfected, cells indicates that interferon is indeed being produced. However, since the induction of both interferon and RIG-I in response to viral infection are RIG-I-dependent and IRF3-dependent events, RIG-I should be up-regulated in the deltaNS1-infected cells. The text is confusing at this point since it introduces a second virus that also lacks a polymerase-associated gene. The logic behind the need for this escaped me, so some clarification of this part of the text would be helpful.

Response: We thank Reviewer 1 for the very positive assessment of our study regarding its significance to the longstanding question surrounding the interferon induction by influenza virus.

Rationale of introducing the NS1 and PA-X double deficient virus:

To facilitate the tracking of RIG-I distribution, our initial intention was to upregulate endogenous RIG-I levels in infected cells by using a mutant virus (deltaNS1) that stimulates IFN production, which in turn may enhance RIG-I expression via the type I IFN autocrine loop (as RIG-I itself is an ISG). However, upregulation was only observed in uninfected neighboring cells via a paracrine signaling, but not in the infected cells despite that these are the cells producing IFN. Since influenza A virus (IAV) PA-X protein is recently reported to have the global host shutoff activity [Khapersky and McCormick, J Virol. 2015 Jul ;89(13):6528-31], we thus constructed the NS1/PA-X double-deficient virus, hoping to ensure that the viral IFN antagonists' functions to be suppressed. Again, no RIG-I upregulation in infected cells was observed. This suggests that IAV-induced host shutoff may not be the cause of unchanged RIG-I levels in infected cells.

RIG-I is upregulated in neighbouring, uninfected cells, but not in deletion virus infected cells:

One possible explanation would be that the IFN signaling axis, rather than RIG-I-mediated IFN induction, is blocked in infected cells, so that the expression of ISGs, including RIG-I, is ceased. This blockage is also independent of viral NS1 and PA-X proteins, as infection with the double-deficient virus did not rescue the inhibitory phenotype. Interestingly, in a previous study, it was demonstrated that IAV infection suppressed STAT1 activation induced by exogenous type I IFNs. The accumulation of viral 5' triphosphate RNA, but not viral proteins, induced NF-kappaB-mediated upregulation of the suppressor of cytokine signaling-3 (SOCS-3) protein, thereby negatively regulating the IFN-JAK-STAT signaling axis and suppressing ISG expression [Pauli et al., PLoS Pathog. 2008 Nov;4(11)] in infected cells. This study thus provides a plausible mechanistic understanding of the unchanged RIG-I levels in IAV-infected cells as observed in our study.

Comments: 2. In lines 199/200 it is argued that early induction of IRF3 by Sendai virus argues against the presence of DI particles. Having worked on Sendai virus for about 35 years I would say it argues strongly for the presence of DI particles! The authors are using the Cantell strain, a strain selected as a good inducer of interferon because it has a propensity to generate DI particles. However, since I cannot see how it matters for these experiments what kind of inducer/PAMP the Sendai virus is making the sentence on DI particles should just be removed.

Response: We completely agree with the reviewer on the robust generation of DI genomes from the Cantell strain of SeV. We believe there was a misunderstanding of the sentence in lines 199/200 read “Of note, compared to IAV infection, SeV infection stimulated IRF3 nuclear translocation as early as 1 h.p.i (Fig. 1a), arguing against the presence of DI RNA in our IAV virus stocks”. Here we tried to compare the kinetics of IRF3 activation induced by SeV with that of IAV. Since SeV Cantell is rich in DI and induced IRF3 nuclear translocation as early as 1 h.p.i, whereas our mutant IAV strains induced IRF3 activation by 4 h.p.i coinciding with the onset of viral replication (Fig. 2a-c), we considered our IAV stocks to be less or free of DI. We now realized that this might not be a proper comparison as the modes of genome replication between the two viruses are different, and the mechanisms of IFN stimulation by the DI genomes of the two viruses might differ too. As suggested by the reviewer we removed this sentence from the paragraph.

Reviewer #2 (Remarks to the Author):

Comments: *Key results: Please summarise what you consider to be the outstanding features of the work. This manuscript reported that nucleus-resident RIG-I senses nonself RNA and activates the MAVS-dependent signaling cascade to induce IFN expression. An elegant experimental design entailing NLS and NES was used to discern the distinct function of nuclear and cytoplasmic RIG-I in sensing Influenza A virus (nuclear) and Sendai virus (cytoplasmic). A novel signaling model was also proposed, which could be significant for related studies in antiviral immunology. These findings are important to advance our understanding in the antiviral activity of RIG-I. However, there are a few major concerns that the author may consider to address.*

Response: We thank reviewer 2 for the appreciation of the significance of our study to the field of antiviral immunity and the positive feedback concerning the experimental design.

Comments: *The amount of nuclear of RIG-I, especially when compared to the cytoplasmic RIG-I, is relatively small or minute. Nuclear RIG-I was more apparent at later time points during viral infection, likely due to further induction by IFN. This raises the question whether cytoplasmic RIG-I is sufficient or nuclear RIG-I is necessary for sensing viral RNA, such as those of IAV. Apparently, the authors have supplied a great deal of data to support their conclusions. One major concern is whether the “reconstituted” expression level of nuclear RIG-I is physiologically comparable to the amount of nuclear endogenous RIG-I of virus-infected cells. Such comparison may be crucial to consolidate the authors’ conclusion.*

Given the concept that nuclear RIG-I signals through the mitochondrion-localized MAVS, it is conceivable that nuclear RIG-I shuttles into the cytoplasm during IAV infection. This data will be important to further validate the role of a nucleus-localized RIG-I in sensing IAV RNA.

The authors rely on IFN reporter assay for the antiviral immune response downstream of RIG-I. When necessary, key experiments have to be validated by real-time PCR analysis of key effectors downstream of the RIG-I signaling pathway.

Response: We thank the reviewer for these constructive comments and we appreciate the importance of a functional comparison between the reconstituted (or induced) expression and endogenous levels of nuclear RIG-I. We conducted the following experiments (Supplementary Fig. 7) to address these specific questions.

Whether the “reconstituted” expression level of nuclear RIG-I is physiologically comparable to the amount of nuclear endogenous RIG-I of virus-infected cells?

New data are presented in Supplementary Fig 7, and line 287-293.

First, in an attempt to obtain the levels of reconstituted nuclear RIG-I physiologically comparable to the endogenous nuclear RIG-I levels, we titrated the doxycycline concentrations on the A549 *RIG-I* KO cell line inducibly expressing NLS-RIG-I (hereafter referred to as iA549-NLS), and proportionally matched the levels of NLS-RIG-I expression to that of the endogenous RIG-I in the nuclear fractions of mock-, WT-, or deltaNS1 virus infected A549 cells. These results were presented in Supplementary Fig. 7a. Note that the NLS-RIG-I expressed in the iA549-NLS cells contains a tandem FLAG tag and a NLS so that it migrated higher than the native RIG-I (when probed with RIG-I antibody). We found that iA549-NLS cells induced by

Dox at 0.005, 0.01, and 0.05 $\mu\text{g/ml}$ gave rise to the comparable or slightly higher levels of nuclear RIG-I expression than the endogenous levels.

Next, to see if this level of nuclear RIG-I is responsive to IAV infection and could induce IFN production, cells were induced for 4 h at these three doses, followed by either mock or deltaNS1 virus (MOI of 1) infection for 14 h (same condition as being used in Fig. 5a and 5b). The qPCR analysis revealed a Dox dose (reflecting RIG-I expression level)-dependent induction of IFN β and IP10 mRNAs upon infection as compared to the “non-induced with infection (-Dox and +virus)” control (Supplementary Fig. 7b, c). Dox induction of nuclear RIG-I expression in the absence of virus infection showed negligible IFN β and IP10 induction (Supplementary Fig. 7b, c). Noticeably, we found that the “non-induced with infection (-Dox and +virus)” control, albeit set to 1 for the $2^{-\Delta\Delta\text{Ct}}$ analysis, exhibited higher levels of antiviral gene expression than the “induced without infection (+Dox and -virus)” conditions. This was likely due to the leaky expression of trace amount of nuclear RIG-I in iA549-NLS cells as shown in the Dox titration experiment (Supplementary Fig. 7a), which could still sense IAV infection. We therefore compared IFN β and IP10 induction in non-induced iA549-NLS cells (without Dox) with that in *RIG-I* KO cells upon infection with deltaNS1 virus. Strikingly, the leaky expression of nuclear RIG-I in non-induced iA549-NLS cells resulted in ~ 35 -fold increase in IFN β mRNA induction compared to *RIG-I* KO cells infected with deltaNS1 virus (Supplementary Fig. 7d). In comparison, no significant induction of IFN β and IP10 mRNA was seen under the same condition for SeV infection (Supplementary Fig. 7d, e), supporting the compartment-specific role for nuclear RIG-I as demonstrated in Fig. 6a-c. Moreover, a Dox dose-dependent reduction of viral mRNA was observed for IAV, but not SeV (Supplementary Fig. 7f, g). Taken together, these additional results consolidate our overexpression experiments demonstrating the antiviral effect of nuclear RIG-I at physiologically relevant levels.

Nuclear RIG-I translocation during IAV infection:

It is clear that the NLS-RIG-I depends on MAVS for downstream signaling transduction (Fig. 4f-h); however, whether the nuclear RIG-I has to translocate from the nucleus to the cytoplasm for downstream adaptor association, and how this translocation is achieved, remains an interesting area for future research. Throughout the immunofluorescence analyses in our study, no redistribution of NLS-tagged RIG-I was seen during the course of IAV infection, yet it mediated efficient IRF3 activation (Fig. 4a, 4h, and Supplementary Fig. 9b). We also monitored the localization of MAVS during infection, which displayed no reorganization but perinuclear distribution (Supplementary Fig. 9b). These results led us to propose the model in which nuclear RIG-I may relay the signal at the nuclear membrane interface without having to shuttle out of the nucleus, particularly given that IAV infection alters the integrity of the nuclear envelop as discussed in the Discussion.

Validation of IFN reporter assay by real-time PCR

New results are presented in Supplementary Fig. 8 and line 301-302 and 349-351.

As suggested by the reviewer, we validated our luciferase reporter assays employed in Fig. 3 and 4a by qPCR for IFN β and IP10 mRNA levels. IAV RNP reconstitution (NA segment) in the presence of SLN-, NES-, or NLS-RIG-I significantly induced IFN β and IP10 mRNA expression as compared to the GFP control. The pattern of differential sensing capacity of SLN-, NES-, and

NLS-RIG-I as determined by luciferase reporter activation is perfectly in line with that determined by qPCR quantitation of IFN β and IP10 mRNA levels.

Comments:

Validity: Does the manuscript have flaws which should prohibit its publication? If so, please provide details.

There are a few major concerns that the authors need to address, please see main summary.

Originality and significance: If the conclusions are not original, please provide relevant references. On a more subjective note, do you feel that the results presented are of immediate interest to many people in your own discipline, and/or to people from several disciplines?

Yes.

Data & methodology: Please comment on the validity of the approach, quality of the data and quality of presentation. Please note that we expect our reviewers to review all data, including any extended data and supplementary information. Is the reporting of data and methodology sufficiently detailed and transparent to enable reproducing the results?

The experiments are well and cleverly designed. Their results are clear-cut and interpreted properly.

Appropriate use of statistics and treatment of uncertainties: All error bars should be defined in the corresponding figure legends; please comment if that's not the case. Please include in your report a specific comment on the appropriateness of any statistical tests, and the accuracy of the description of any error bars and probability values.

Scientific statistical analysis is well performed, except the correlation results. It would be better to show the correlation coefficient or its equivalent to support the positive/negative correlation.

Response: We have removed unnecessary descriptions for positive/negative correlation except that for the colocalization analysis.

Comments: *Conclusions: Do you find that the conclusions and data interpretation are robust, valid and reliable?*

Yes, the authors' conclusions are generally supported by their experimental results. However, the function of nuclear RIG-I may have to be better studied with a controlled expression comparable to the endogenous level of nuclear RIG-I.

Suggested improvements: Please list additional experiments or data that could help strengthening the work in a revision.

Please see comments in the summary that protein expression level may be better controlled.

Response: These are all addressed above.

Comments: *References: Does this manuscript reference previous literature appropriately? If not, what references should be included or excluded?*

Yes.

Clarity and context: Is the abstract clear, accessible? Are abstract, introduction and conclusions appropriate?

Introduction: Introduction should be condensed to improve the readability. For example, the first two paragraphs can be combined to one that focuses on the RNA sensing studies.

Response: We have combined the first two paragraphs in the Introduction as suggested to emphasize the RNA sensing studies.

Comments: Line 83: Please use most-studied rather than “best-studied”.

Response: Corrected as suggested.

Comments: Line 95: Please define “vRNP” at the first place.

Response: vRNP was defined.

Comments: Results: For SLN- and NES experiments, why use doxycycline as stimuli, not IFN β stimulation or SeV (this is used to induce RIG-I expression)? In line 154-155, the IAV infection (also refer to Figure 1E) is a sequential experiment, need an explanation.

Response: The A549 cell lines expressing SLN- or NES-RIG-I were constructed in the genetic background of RIG-I knockout to avoid interference from endogenous RIG-I. These cells have lost endogenous RIG-I and the expression of SLN- or NES-RIG-I in these cell lines was independently controlled from the complementing episomal constructs driven by a doxycycline-inducible Tet-on promoter rather than the natural RIG-I promoter induced by SeV infection or IFN priming.

We have provided additional description for Fig. 1e in the text (line 147-148) to clarify the intention of this experiment.

Comments: Line 120: ...we subjected A549 cells to SeV infection or IFN β stimulation... to correspond the order of following description.

Response: Corrected.

Comments: Line 127: Please define “IRF3”.

Response: IRF3 is defined.

Comments: Figure 2: Please be consistent with the color coatings. Use same color to represent the IRF3 throughout the figure or manuscript (if possible?).

Response: We thank the reviewer for this suggestion on the color code consistency. The color codes were chosen based on the Alexa fluorophores conjugated to the secondary antibodies in use, except for the images scanned with four channels, where one was picked arbitrarily (usually the Alexa 546). In this revised manuscript, the red channel was assigned to IRF3 in all experiments scanned with three channels, as it achieves the highest level of contrast for visualization when merging with the green channel. In all the other experiments scanned with four channels, cyan was selected for IRF3 as the red channel serves better clarity for viral RNA visualization.

Comments: The RIG-I expression in Δ NS1-ms infected condition was not apparent at 8 h.p.i, but it was quite distinct at 14 h.p.i. comparing to mock-infected group in the Supplementary Fig. 3A. Please revise the statement in Line 169-171 and following data interpretation. Because of the life cycle?

Response: We have revised the text accordingly to better emphasize the observations from the single-cycle vs. prolonged infection with IAV. The take home message is that RIG-I expression

was not upregulated in the infected cells, regardless of the time course of infection. RIG-I upregulation was only seen in the uninfected neighboring cells at later time points post-infection through a paracrine signaling. This was addressed in our response to reviewer 1.

Comments: Line 201-203: *The statement claims the two mutants indeed activate RIG-I, however contradict to the earlier description in Line 166-171.*

Response: Indeed, line 166-171 states that endogenous RIG-I remained at basal levels in IAV-infected cells during either a single-cycle or prolonged infection. However, this basal expression of RIG-I was active and was sufficient for IAV sensing and mounting an interferon response in the infected cells, as demonstrated by the efficient IRF3 nuclear translocation in infected cells (Fig. 2a, b). Furthermore, the IFN produced from infected cells was sufficient for upregulation of RIG-I in uninfected neighboring cells (Supplementary Fig. 3a). We consider this level of RIG-I activity is “efficient” as conveyed in line 201-201 (initial version). Nevertheless, to minimize the confusion, we changed “RIG-I activation” to “IRF3 nuclear translocation” (line 192-193).

Comments: Line 233-235: *Please describe the reconstitution system in the Methods.*

Response: This information was provided. Please refer to lines 686-700 in the initial version under the section of “RNP reconstitution and luciferase reporter assay” in the Methods for details.

Comments: Line 253-254: *Please also perform the statistical analysis, supporting the NLS-RIG-I inactive. Also, the y-axle in Figure C and D are 20 times difference, how could compare them in a comparable way?*

Response: The statistical analysis was conducted as suggested for Fig. 3c and no statistical significance in IFN promoter activation was seen between the mock- and 5’ppp-RNA transfected cells expressing NLS-RIG-I, confirming that NLS-RIG-I was inactive in sensing cytoplasmic ligands. The statistical analysis results have now been included into the corresponding figure legend.

Fig 3c and 3d were done using different RNA ligands. In Fig. 3c, cells were transfected with synthetic 5’ppp-dsRNA (Invivogen), which is a 19-bp fully complementary dsRNA. Upon transfection in nanogram-scale, it serves as a potent RIG-I agonist. In Fig. 3d, the RIG-I ligands are derived from reconstituted influenza like vRNAs. The panhandle structure of IAV-RNA genome, which acts as RIG-I agonist, is only partially complementary and thus is not as potent as a fully complementary ligand [J Virol. 2015 Jun;89(11):6067-79]. The fold difference between Fig. 3c and 3d likely reflects the potency of the immunostimulatory RNA in action and/or the amount of the panhandle structure of IAV-derived RIG-I agonists. We do not compare the results between these two experimental settings. More importantly, compared to other compartment-specific RIG-I, nuclear RIG-I did not respond to cytoplasmic RIG-I ligand in Fig. 3c; whereas in response to the reconstituted viral RNA, which mimics the RNA ligands during IAV infection, both nuclear RIG-I (Fig. 3e) and SLN-tagged RIG-I are activated (Fig. 3d).

Comments: Line 277-280: *When authors claim the positive/negative correlation, could authors also provide quantitative results, such as correlation coefficient or similar indicators.*

Response: As mentioned above, we have rephrased the text to remove unnecessary descriptions for positive/negative correlation except that for the colocalization analysis.

***Comments:** I wonder whether the expression of p-IRF3 and IRF3 in cytoplasmic and nucleus fractions in NES and NLS differ? In particular, to explore the “cooperative IFN induction along with its cytoplasmic counterpart”.*

Response: IRF3 locates in the cytoplasm in resting state and is translocated into the nucleus upon phosphorylation by upstream kinases following the activation of signaling pathways, such as the RIG-I pathway. The activation of either NES or NLS-RIG-I signaling will lead to the nuclear translocation of pIRF3. We did monitor the levels of IRF3 phosphorylation (pIRF3) in infected cells expressing NES- or NLS-RIG-I, and demonstrated that both of them gave rise to delayed IRF3 activation (Fig. 4b).

***Comments:** Line 360: ...across infected cell population? Please clarify.*

Response: Although multiple scanning fields were taken in the immunofluorescence analysis for the quantification of IRF3 nuclear translocation, they only represented some, but not all of the cells. We further analyzed the levels of IRF3 phosphorylation in the infected population by Western blotting to reinforce that NLS-RIG-I was inactive in sensing cytoplasmic-replicating SeV.

***Comments:** Discussion: Please consider remove the first paragraph in the Discussion, which is redundant, repeating the description of abstract and the part of the introduction.*

Response: As suggested, we have removed the redundant sentences from the first paragraph in the Discussion but kept the key text highlighting the significance of our study.

We hope we addressed the reviewers' comments properly and these revisions make this manuscript acceptable for publishing in the Nature communications.

Sincerely yours,

Corresponding author:

Yan Zhou, Ph.D.

Vaccine and Infectious Disease Organization

University of Saskatchewan, Saskatoon

Saskatchewan S7N 5E3 Canada

REVIEWERS' COMMENTS:

Reviewer #1 (Remarks to the Author):

This is a revised manuscript which has made significant efforts to address both reviewers' comments.

Reviewer 1 had asked why RIG-I levels did not appear to be induced in the infected cells given that the virus used lacking NS1. The authors have offered an explanation for this, and for why they used a virus that was also deficient in PA-X, and this is now adequately addressed in the text.

Reviewer 1 asked for the confusion about the presence of Sendai virus DI particles to be addressed, and the authors have now done this.

Reviewer 2 raised a number of text issues, all of which have been addressed. Reviewer 2 also raised three technical questions, and the authors have made a reasonable job of addressing these, providing additional data. Specifically, reviewer 2 queried whether the levels of ectopically expressed RIG-I that rescued IFN inducibility were artefactually high; the authors have carried out a titration experiment using an inducible promoter to generate a concentration gradient of RIG-I, and have shown that even very low levels of ectopically expressed nuclear RIG-I (which are equivalent to the normally very low levels of RIG-I) are sufficient for IFN induction in response to IAV. Reviewer 2 also raised the interesting issue of how nuclear RIG-I could signal to cytoplasmic MAVS. The reviewer did not ask for experiments on this and agreed that this was a subject for future research, but the authors have noted that they have not observed redistribution of nuclear RIG-I or cytoplasmic MAVS during IAV infection. Reviewer 2 also asked that the luciferase reporter data that supported the induction of the IFN promoter should be validated by qPCR, which the authors have convincingly done.

Both reviewers commented in their original reports that this was an interesting and innovative manuscript, and that the experiments were well conceived. The revised manuscript addresses all of the comments.